# Memory-like states created by the first ethanol experience are encoded into the *Drosophila* mushroom body learning and memory circuitry in an ethanol-specific manner

**Caleb Larnerd[1], Maria Nolazco[2], Ashley Valdez[2¤], Vanessa Sanchez[1], Fred W. Wolf[1,3]***

**1** Quantitative and Systems Biology Graduate Program, University of California, Merced, California, United States of America, **2** Biological Sciences Undergraduate Program, University of California, Merced, California, United States of America, **3** Department of Molecular and Cell Biology, University of California, Merced, California, United States of America

¤ Current address: Neuroscience Graduate Program, University of California, Riverside, United States of America
* fwolf@ucmerced.edu

## Abstract

A first ethanol exposure creates three memory-like states in *Drosophila*. Ethanol memory-like states appear genetically and behaviorally paralleled to the canonical learning and memory traces anesthesia-sensitive, anesthesia-resistant, and long-term memory ASM, ARM, and LTM. It is unknown if these ethanol memory-like states are also encoded by the canonical learning and memory circuitry that is centered on the mushroom bodies. We show that the three ethanol memory-like states, anesthesia-sensitive tolerance (AST) and anesthesia resistant tolerance (ART) created by ethanol sedation to a moderately high ethanol exposure, and chronic tolerance created by a longer low concentration ethanol exposure, each engage the mushroom body circuitry differently. Moreover, critical encoding steps for ethanol memory-like states reside outside the mushroom body circuitry, and within the mushroom body circuitry they are markedly distinct from classical memory traces. Thus, the first ethanol exposure creates distinct memory-like states in ethanol-specific circuits and impacts the function of learning and memory circuitry in ways that might influence the formation and retention of other memories.

## Author summary

The first alcohol experience creates multiple distinct memory-like states in *Drosophila* that depend on alcohol dose and exposure length. These alcohol memory-like states are likely acted on by subsequent intake to cause more persistent changes in brain function. Because the three alcohol memory-like states mirror classical associative memories, we investigated alcohol encoding mechanisms in the associative circuitry. We discovered that, while there is overlap for all three alcohol memories, it is most extensive for the

**Data availability statement:** All data is included in the manuscript and its associated files.

**Funding:** This research was supported by National Institutes of Health (NIH) National Institute on Alcohol Abuse and Alcoholism (NIAAA) grants AA028352 and AA029178 to F.W.W. and by the National Science Foundation (NSF) Graduate Research Fellowship Program to C.L. The funders had no role in study design, data collection and analysis, decision to publish, or preparation of the manuscript.

**Competing interests:** The authors have declared that no competing interests exist.

shortest form that is labile, and less so for the two consolidated forms. Partial use of classical associative circuitry points to the existence of alcohol-specific memory circuits and a means for alcohol to modify classical associative memories.

## Introduction

Drugs of abuse create strong and lasting memories that persist long after drug use has ceased. The unusual persistence and the nature of drug memories are thought to underlie the high risk for relapse into drug use after periods of abstinence. A longstanding question is how drug memories are related to other forms of memory. Human and rodent addiction studies suggest dense overlap of mechanisms at the molecular level, and partial overlap at the circuit level [1–3]. A major challenge is to identify what makes memories of drugs of abuse unique. Ethanol, the active ingredient of alcohol, causes plasticity at the molecular, synaptic, and circuit levels. The prevalence of alcohol use disorder and its costs to society increase the value of understanding its root mechanisms of action [4].

Mammalian brains are complex and the mechanisms for learning and memory are diverse. Organisms with simpler brains are useful for identifying the molecular basis of memory at the level of individual neurons and in the context of complete circuits. The brain of the fly *Drosophila melanogaster* has about 1 million-fold fewer neurons than the human brain, and yet flies form multiple separable types of memories that have specific valences, persistence, and associations. The *Drosophila* mushroom bodies are the major brain center for associative memories, including their acquisition, consolidation, and expression. The circuitry for the mushroom bodies is completely described and their encoding of learning and memory in the circuitry is known in great detail [5].

*Drosophila* is a useful model for understanding how ethanol affects the brain, including the contribution of ethanol memories [6]. Spaced training to associate neutral cues with ethanol inebriation forms long lasting positive valence memories that require the mushroom bodies for acquisition, consolidation, and expression [7–9]. More recently it was discovered that a single ethanol exposure given to ethanol-naive flies without an explicit associative pairing also creates memory-like states [10]. A highly inebriating ethanol exposure creates a labile and a consolidated memory-like state. A longer low dose of ethanol creates a third, independent long-term memory-like state that requires protein synthesis and the CREB transcriptional regulator in the brain [10,11]. These three single exposure memory-like states are the basis for rapid and chronic ethanol tolerance development, where tolerance is defined as acquired resistance to the inebriating and sedating actions of ethanol. Rapid tolerance is operationally defined as occurring after internal ethanol concentrations are returned to baseline from the initial inebriating exposure [12,13]. Both rapid and chronic tolerance in *Drosophila* are due to functional changes in the nervous system and not changes in ethanol metabolism [11,13].

Rapid tolerance was first defined in vertebrates, where most major neurotransmitters, a number of neuromodulators and hormones, as well as transcriptional regulatory mechanisms are implicated [12,14,15]. The role of learning and memory in tolerance is less studied. In classical studies with rodents, an inebriating dose of ethanol was paired with a difficult coordination task: the animals performed badly for the first dose and better during a second tolerance testing dose; the performance improvement was dependent on new protein synthesis [16–18]. Animals given the task only during the tolerance testing dose performed no better than the first dose animals. Single ethanol exposure memory-like states are candidate substrates that are acted upon by subsequent ethanol exposures to create longer lasting memories and addiction-like states.

How single ethanol exposure memory-like states are encoded in the brain is not known. Here, we focus on the genes and circuitry defined for classical aversive learning and memory, because it is currently better defined in *Drosophila* as compared to appetitive learning and memory, and because ethanol has both aversive and appetitive properties within a single exposure [7,19–21]. We find that ethanol memory-like states correspond molecularly and behaviorally to the classically conditioned memory traces of anesthesia-sensitive memory (ASM), anesthesia-resistant memory (ARM), and long-term memory (LTM). Hence, we name them anesthesia-sensitive tolerance (AST), anesthesia-resistant tolerance (ART), and chronic tolerance. We asked if single ethanol exposure memory-like states are encoded into the mushroom body circuitry similarly to their cognate classically conditioned memories. We focused on the role of three critical neuron types in the mushroom body circuitry: the mushroom body intrinsic Kenyon cells, the Anterior Paired Lateral (APL) neurons, and the Dorsal Paired Medial (DPM) neurons; memory encoding by these neurons can point to directed experiments to test the functionally more complicated other circuit elements. Approximately 2000–2500 mushroom body-intrinsic Kenyon cells with highly parallelized fibers form three distinct lobes, the αβ, α'β', and the γ lobes [22]. The Kenyon cells receive inputs from sensory pathways for the conditioned stimulus, twenty types of dopaminergic neurons that convey internal state and unconditioned stimulus information, and two very large neurons—the APL and the DPM – that broadly ramify across the Kenyon cells and that perform multiple functions [23–27]. The Kenyon cells send output to a small number of output neurons (MBONs) that encode either positive (approach) or negative (avoidance) behavioral responses [8]. Classical associative learning biases Kenyon cell-to-MBON synapses to favor either approach or avoidance to subsequent presentation of the conditioned stimulus [28]. Our findings indicate that single exposure memory-like states are encoded into the mushroom body circuitry differently from classical learning and memory, but with partial overlap. Moreover, single exposure memory-like state circuits are also distinct from memories created by spaced training to associate a neutral cue with ethanol inebriation [7,9,29]. We speculate that partial overlap of genes and circuits for ethanol and classical memories may provide a mechanism for ethanol to impact other memory mechanisms.

## Results

### A single sedating dose of ethanol forms AST and ART

Two distinct memory-like states contribute to rapid ethanol tolerance [10]. Here, we characterize their properties. Rapid tolerance is induced and measured by giving flies two ethanol exposures of identical concentration, with the second exposure (E2) beginning 4–24 hr after the start of the first (E1) [30]. The number of flies in a group of 20 genetically identical individuals that lose the righting response are counted every 6 min, and the time to 50% sedation (ST50) for the group is calculated for E1 and E2. Sensitivity is the ST50 for E1, and rapid tolerance is the ST50, E2-E1. A sedating dose of ethanol created rapid tolerance that was diminished by cold shock anesthesia 30 min or 3 hr after cessation of the initiating dose (Figs 1A and S1A). Rapid tolerance remained detectable at 24 hr, but 24 hr tolerance was insensitive to cold shocks 30 min or 23 hr after cessation of the initiating dose (Figs 1A and S1A). Thus, labile AST is dissipated between 3 and 24 hr, and tolerance measured at 24 hr is solely composed of consolidated ART. The temporal properties of AST and ART are consistent with those of aversively conditioned ASM and ARM [31].

Cold shock-induced deficits in rapid tolerance may arise in part from non-specific disruptions of neural activity. As a second test for AST, we investigated the role of the neuropeptide *amnesiac* (*amn*), that is necessary and sufficient specifically for aversively-conditioned ASM

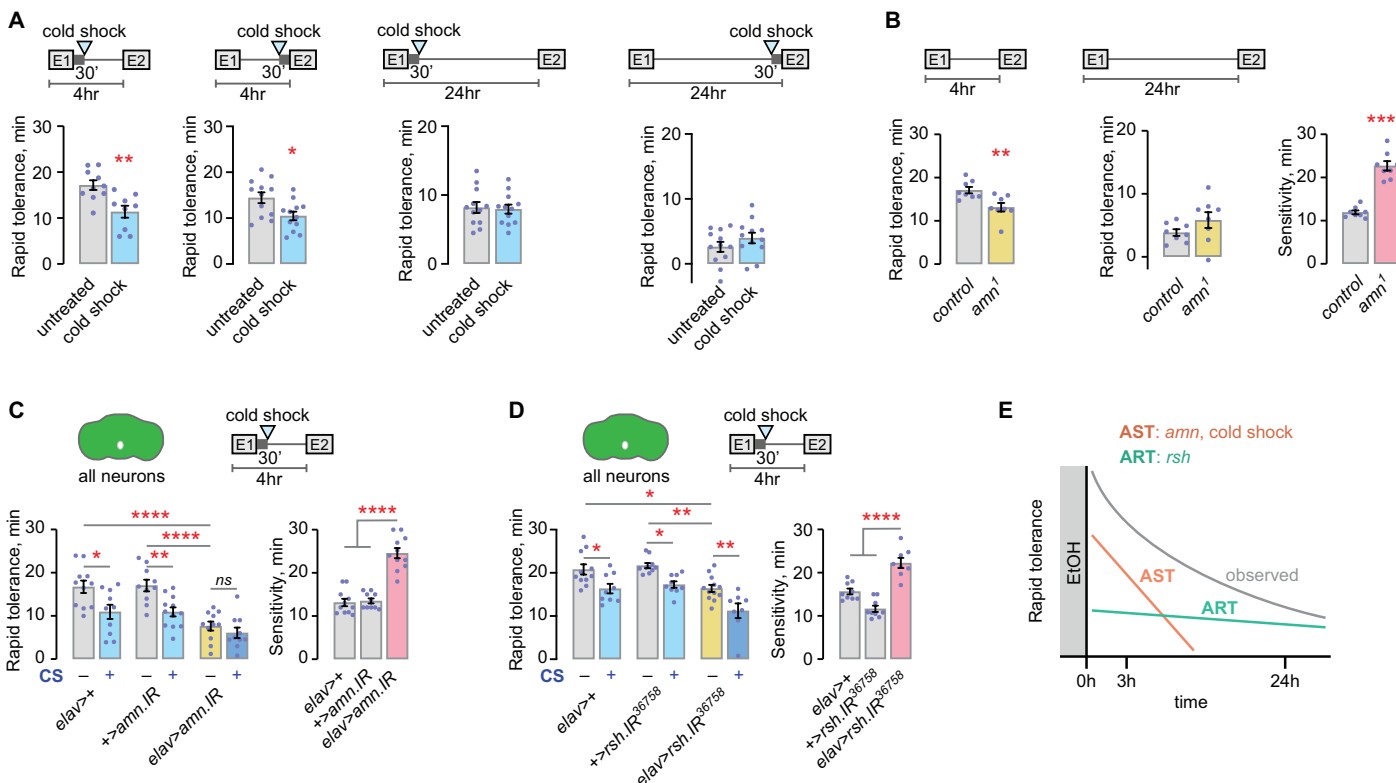

**Fig 1. An acute, sedating dose of ethanol forms separable AST and ART. A**) Cold shock anesthesia decreases 4 hr but not 24 hr rapid tolerance. Left and center-left: Rapid tolerance is cold shock sensitive for at least 3 hr after acute ethanol exposure. Center-right and right: rapid tolerance measured at 24 hr is cold-shock insensitive. Unpaired t test (two-tailed). **B**) *amn* promotes AST and sedation sensitivity, but not ART. Left: unpaired t test (two-tailed); right: Welch's t test (two-tailed). Sensitivity: Welch's t test (two-tailed). **C**) Neuronal *amn* promotes AST and sedation sensitivity. Tolerance, left: one-way ANOVA/Holm-Šídák's. Sensitivity, right: one-way ANOVA/Holm-Šídák's. **D**) Neuronal *rsh* supports ART and sedation sensitivity. Tolerance, left: one-way ANOVA/Holm-Šídák's. Sensitivity: one-way ANOVA/Holm-Šídák's. **E**) Neuronally encoded AST and ART dynamics for rapid tolerance encoding.

[32–35]. *amn¹* mutant flies display reduced rapid tolerance at 4 hr after the initiating dose, but not at 24 hr (Fig 1B), suggesting that *amn*-dependent genetic pathways support AST. *amn¹* mutants were also resistant to ethanol sedation (Fig 1B). Amn is a predicted activator of adenylyl cyclase (AC), that generates cAMP and is encoded by the *rutabaga* (*rut*) gene in *Drosophila*. *rut* supports coincidence detection for associative learning paradigms that pair conditioned and unconditioned stimuli [36,37]. In our assay, *rut¹* mutants displayed a rapid tolerance deficit at 4 hr, not at 24 hr, and they show reduced sedation sensitivity (S1B Fig). Thus, *rut*, like *amn*, specifically functions in AST, similar to its role in ASM olfactory conditioning. Pan-neuronal expression of an RNAi against *amn* reduced rapid tolerance and sensitivity (Fig 1C). Adding a cold shock 30 min after the inducing exposure reduced tolerance in controls, but not in the experimental, suggesting that neuronal *amn* acts in the same pathway as cold shock anesthesia, and does so in the nervous system.

Next, we tested for neuronal localization of ART by decreasing expression of *rsh*, a gene that is specifically required for ARM in classical olfactory conditioning [38–41]. Pan-neuronal expression of an RNAi against *rsh* reduced rapid tolerance and sensitivity (Fig 1D). Adding a cold shock 30 min after the inducing exposure reduced tolerance in both the experimental flies (*elav>rsh. IR*) and in the genetic controls, in contrast to *amn*. A second RNAi against *rsh* also reduced rapid tolerance when expressed in all neurons (S1C Fig). Thus, AST and ART are genetically and

behaviorally distinct: rapid tolerance is composed of quickly decaying labile AST that requires neuronal *amn*, and longer lasting consolidated ART that requires neuronal *rsh* (Fig 1E).

## Labile AST is amn-dependent in the mushroom body Kenyon cells

All subtypes of Kenyon cells and the mushroom body-extrinsic DPM neurons express *amn* [35,42–44]. RNAi against *amn* in all mushroom body Kenyon cells reduced rapid tolerance (Fig 2A). Moreover, a cold shock did not further decrease rapid tolerance in flies lacking *amn* in the Kenyon cells, indicating that cold shock anesthesia works through an *amn* pathway in Kenyon cells, as it does for ASM (Fig 2A) [43]. Conversely, *amn* knockdown in the DPMs did not affect rapid tolerance (Fig 2B). Thus, rapid tolerance requires *amn* in the Kenyon cells and not the DPMs, similar to aversively conditioned ASM. To localize its role in rapid tolerance, we decreased *amn* expression in each of the three Kenyon cell groups that form the three anatomical lobes of the mushroom bodies, the αβ, α'β', and γ lobes. *amn* was specifically required in the γ lobe to promote rapid tolerance (Figs 2C and S2A). Thus, labile AST requires *amn* in the γ lobes, in contrast to aversively conditioned ASM that requires *amn* in the αβ lobes [43].

Aversive ASM but not ARM requires glutamatergic NMDA receptors in the mushroom body Kenyon cells, consistent with the role of $Ca^{2+}$-dependent synaptic plasticity in olfactory conditioning [45]. Flies mutant for the NMDA receptor type I subunit *Nmdar1*, or for *dlg1* that encodes an NMDAR scaffolding protein, show decreased rapid tolerance [46]. We used an RNAi against the *Nmdar1* to test its function in the mushroom bodies. *Nmdar1* knockdown in all Kenyon cells decreased rapid tolerance at 4 hr but not at 24 hr, indicating that glutamatergic NMDARs support AST (Fig 2D). Ca2+/calmodulin-dependent protein kinase II (CaMKII) functions downstream of NMDAR for memory formation [47]. RNAi against CaMKII in the mushroom body Kenyon cells did not affect tolerance, whereas pan-neuronal RNAi decreased rapid tolerance (Fig S2B, B'). Thus, *Nmdar1*-dependent signaling is shared between AST and aversive ASM but the signal transduction pathway may be different (Fig 2E).

Ethanol sensitivity mapped in a partially overlapping pattern with rapid tolerance (Fig 2A'-D'). A small but significant role for *amn* was uncovered when *amn* RNAi was expressed specifically in the γ Kenyon cells (Fig 2C'). *Nmdar1* was required for ethanol sensitivity in all Kenyon cells (Fig 2D').

## Radish-dependent ART is mushroom body independent

Genes and circuits in the mushroom bodies support ARM [40]. Considering that neuronal expression of the ARM gene *rsh* supports rapid tolerance (Fig 1D), we hypothesized that mushroom body neurons hold *rsh*-dependent ART. However, targeting Kenyon cells with multiple *rsh* RNAis did not affect rapid tolerance (Figs 3A and S3A). To rule out the possibility that a *rsh* knockdown causes a gain in AST and thus masks a deficit in ART, we also tested the manipulation under conditions that lack AST. Even when AST was blocked by cold shock or dissipated with time, ART remained unaffected by *rsh* knockdown in all Kenyon cells (Figs 3A and S3A). To search for a site of action for *rsh*-dependent rapid tolerance, we next investigated extrinsic mushroom body neurons. *rsh* was dispensable in the APLs for rapid tolerance (Figs 3B and S3B). Similarly, *rsh* was dispensable in the DPMs for rapid tolerance (Fig 3C). Thus, the bilateral pairs of DPM and APL neurons, that each broadly innervate the mushroom bodies (S3C, D Fig), lack coding capacity for *rsh*-dependent ART.

In conclusion, *rsh* supports rapid tolerance in neurons outside the mushroom bodies. Although functional localization of *rsh* has not been well characterized for ARM, a known site of action is in αβ neurons [5,41]. Thus, not only does *rsh*-dependent ART differ from ARM, *rsh*-dependent ART and *amn*-dependent AST arise from separate circuits in the *Drosophila* brain (Fig 3D).

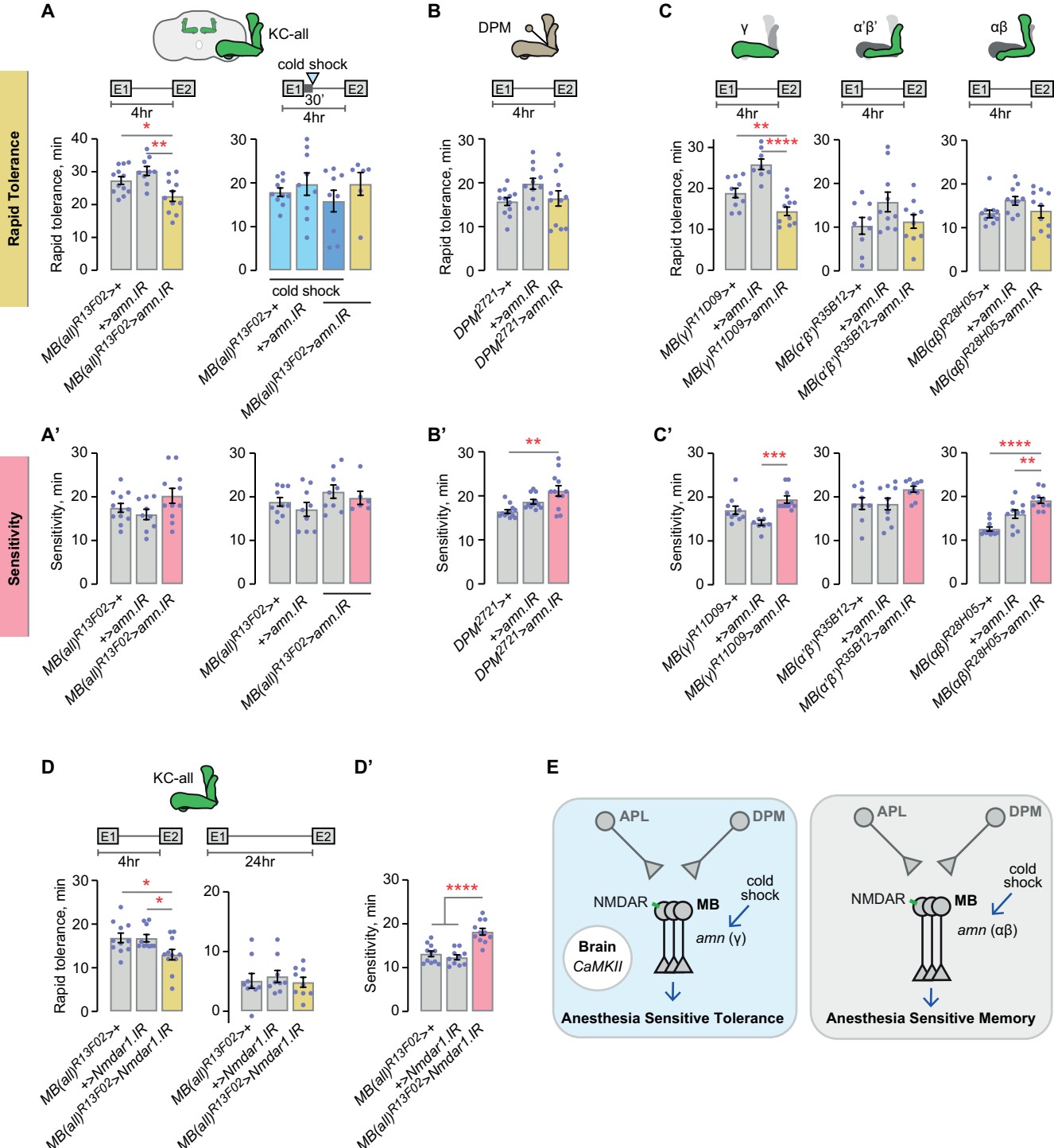

**Fig 2. Amnesiac-dependent AST resides in the mushroom body Kenyon cells. A, A')** *amn* in all mushroom body Kenyon cells promotes rapid tolerance and its effect is sensitive to cold shock, whereas there is no effect on sedation sensitivity. Tolerance, left: one-way ANOVA/Holm-Šídák's; right: one-way ANOVA. Sensitivity, left: one-way ANOVA; right: Kruskal-Wallis. **B, B')** *amn* is dispensable in DPM neurons for rapid tolerance and sedation sensitivity. Tolerance: one-way ANOVA. Sensitivity: Kruskal-Wallis. **C, C')** *amn* in the mushroom body γ lobe Kenyon cells promotes rapid tolerance. *amn* in the mushroom body αβ lobe Kenyon cells promotes sedation sensitivity. Tolerance, left: one-way ANOVA/Holm-Šídák's; center: one-way ANOVA; right: one-way ANOVA. Sensitivity, left: Kruskal-Wallis/Dunn's; center: one-way ANOVA; right: one-way ANOVA/Holm-Šídák's. **D, D')** *Nmdar1* in all mushroom body Kenyon cells promotes AST and sedation sensitivity. Tolerance, left: one-way ANOVA/Holm-Šídák's; right: one-way ANOVA. Sensitivity: one-way ANOVA/Holm-Šídák's. **E)** Summary of AST encoding by anesthesia, NMDAR, and amn in the mushroom body circuitry, as compared to ASM.

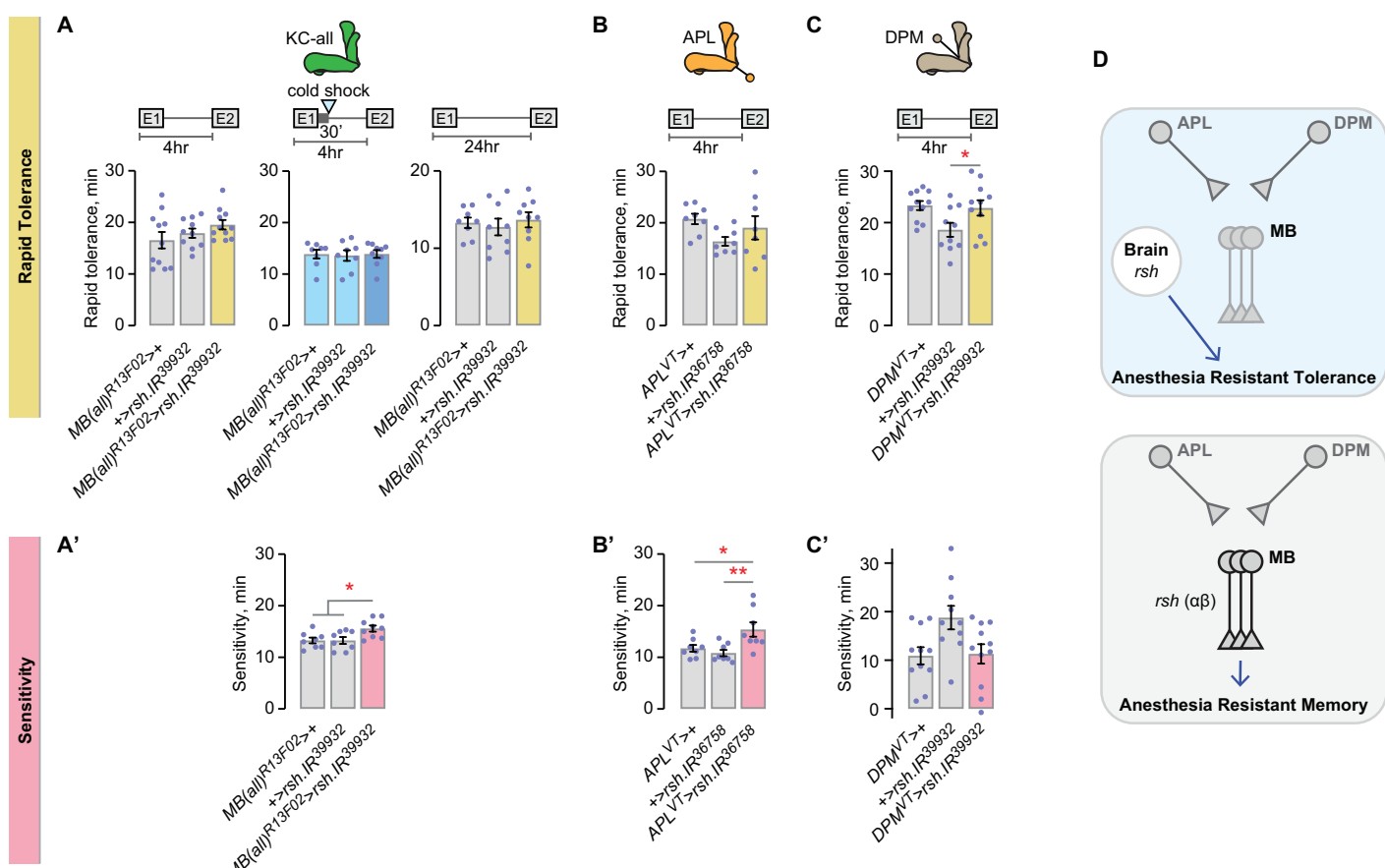

**Fig 3. Radish-dependent ART resides outside the mushroom bodies. A, A'**) *rsh* in the Kenyon cells is dispensable for ART, whereas it promotes sedation sensitivity. Tolerance, left: Brown-Forsythe ANOVA; center: Kruskal-Wallis; right: one-way ANOVA. Sensitivity: one-way ANOVA/Holm-Šídák's. **B, B'**) *rsh* in the APL neurons is dispensable for ART, but it promotes sedation sensitivity. Tolerance: Brown-Forsythe. Sensitivity: one-way ANOVA/Holm-Šídák's. **C, C'**) *rsh* in the DPM neurons is dispensable for rapid tolerance and sedation sensitivity (C'). Tolerance: one-way ANOVA/Holm-Šídák's. Sensitivity: one-way ANOVA. **D**) Radish-dependent encoding of ART, as compared to ARM, in the mushroom body circuitry.

Radish-dependent ethanol sensitivity mapped to the Kenyon cells, in contrast to ART (Figs 3A, S3A'). Radish-dependent ethanol sensitivity may also map to the APL neurons, however decreased sensitivity was only evident with one of two Radish RNAis (Figs 3B', S3B'). Radish did not map to the DPM neurons for ethanol sensitivity (Fig 3C'). Thus, while the role of Radish in ART is encoded outside the mushroom bodies, it's role in ethanol sensitivity includes the Kenyon cells.

## Temporally precise APL activity induces rapid tolerance

The APLs regulate learning and memory via broad innervation of Kenyon cells, providing GABAergic feedback inhibition to support sparse encoding of olfactory stimuli and octopaminergic support of ARM [48–50]. We tested the role of the APLs in controlling ethanol rapid tolerance by inactivating them via expression of temperature-sensitive *Shibire* (*Shi*)—a dominant mutant form of dynamin that blocks vesicular endocytosis and presynaptic release [51]. Inactivating adult APL neurons marked by the *GH146-Gal4* driver decreased rapid tolerance (Fig 4A). Addition of *Gad1-Gal80* removed the APLs from the GH146 expression pattern and the effect on rapid tolerance was abolished (Fig

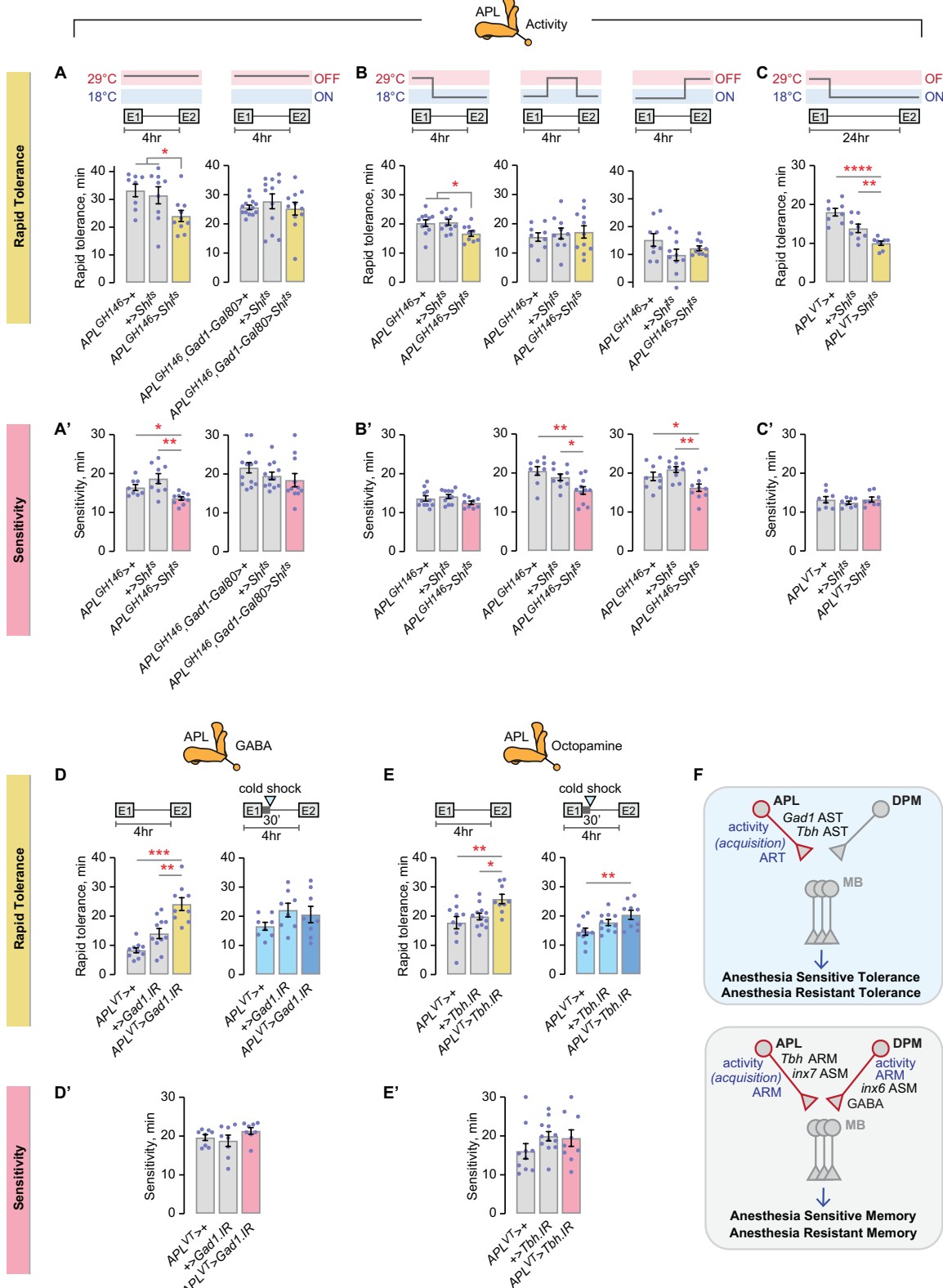

**Fig 4. APL activity during tolerance acquisition promotes rapid tolerance. A, A')** Left: inactivation of the APL neurons through-out the rapid tolerance paradigm decreased rapid tolerance (A), and increased sedation sensitivity (A'). Right: subtraction of GAB-Aergic neurons (*Gad1-Gal80*), including the APL, blocks the effects of neuronal inactivation for both rapid tolerance and sedation

sensitivity. Tolerance, left: one-way ANOVA/Holm-Šídák's; right: Kruskal-Wallis. Sensitivity, left: Brown-Forsythe/Dunnett's; right: Kruskal-Wallis/Dunn's. **B, B'**) Inactivation of the APL neurons in *GH146-Gal4* specifically during tolerance acquisition decreased rapid tolerance (B). Inactivation of neurons in the *GH146-Gal4* pattern during the inter-exposure interval or during the expression phase increased sedation sensitivity (B'). Tolerance, left: one-way ANOVA/Holm-Šídák's; center: one-way ANOVA; right: Brown-Forsythe. Sensitivity, left: one-way ANOVA; center: one-way ANOVA/Holm-Šídák's; right: one-way ANOVA/Holm-Šídák's. **C, C'**) APL inactivation during rapid tolerance acquisition decreased ART measured at 24 hr. Tolerance: one-way ANOVA/Holm-Šídák's. Sensitivity: one-way ANOVA. **D, D'**) RNAi against *Gad1* in the APL neurons increases rapid tolerance (left), and the effect is sensitive to cold shock (right). There is no effect on sedation sensitivity. Tolerance, left: Brown-Forsythe/Dunnett's; right: one-way ANOVA. Sensitivity: one-way ANOVA. **E, E'**) RNAi against *Tbh* in the APL neurons increases rapid tolerance (left), and the effect is sensitive to cold shock (right). There is no effect on sedation sensitivity. Tolerance, left: one-way ANOVA/Holm-Šídák's; right: one-way ANOVA/Holm-Šídák's. Sensitivity: one-way ANOVA. **F**) Activity-dependent encoding of AST and ART, as compared to ASM and ARM.

4A). Moreover, inactivating adult APL neurons using the *VT43924-Gal4* APL driver also decreased rapid tolerance (S4A Fig). Taken together, the APL neurons promote rapid tolerance.

We temporally dissected rapid tolerance into three phases: acquisition (E1, first ethanol exposure), inter-exposure interval, and expression (E2, second ethanol exposure). We found that APL activity was required specifically during rapid tolerance acquisition (Fig 4B). Hence, the APLs regulate the initial development of tolerance. Rapid tolerance measured at 24 hr also required APL activity during the acquisition phase (Fig 4C). The simplest interpretation is that the APLs regulate the acquisition of ART, however we cannot rule out a second role in AST acquisition.

To test for the role of GABA and octopamine release, we expressed in the APLs RNAi against the GABA biosynthetic enzyme Gad1 and the octopamine biosynthetic enzyme Tbh. *Gad1* RNAi in the APLs resulted in increased rapid tolerance that was sensitive to cold shock anesthesia (Figs 4D and S4B). Similarly, *Tbh* RNAi in the APLs increased rapid tolerance that was cold shock sensitive (Figs 4E and S4C). Thus, the APLs constitutively require both GABA and octopamine to limit AST.

We next tested neural activity in the DPM neurons that help consolidate associative learning of aversive stimuli [33,38,52,53]. The adult DPMs were dispensable for rapid tolerance when synaptic release was blocked throughout the rapid tolerance paradigm or specifically during the inter-exposure interval (S4D Fig). Consistent with no role in rapid tolerance, GABA synthesis in the DPMs was dispensable (S4E Fig). Thus, unlike ASM or ARM, rapid tolerance does not require the DPM neurons.

Gap junctions between the APL and DPM neurons are required for aversively-conditioned ASM, via the heterotypic pairing of Innexin 7 in the APL and Innexin 6 in the DPM [54]. We tested APL:DPM gap junctions for a role in rapid tolerance. RNAi reduction of *inx7* in the APLs or *inx6* in the DPMs had no effect on rapid tolerance (S4F Fig). Thus, the APL:DPM Innexin gap junctions that support ASM are dispensable for rapid tolerance.

In conclusion, APL neuronal activity during the initiating ethanol exposure promotes ART, and release of GABA and octopamine limit AST. DPM activity and APL-DPM gap junctions are dispensable. Thus, rapid tolerance has distinct circuit features from ASM and ARM in the APL and DPM giant interneurons (Fig 4F).

APL and DPM neurons encode ethanol sensitivity separably from rapid tolerance. The effect of acute inactivation of the APL neurons on ethanol sensitivity was different for the two *APL-Gal4* drivers used, suggesting that neurons other than the APLs are responsible (Figs 4A'-C', S4A'). No role for APL GABA or octopamine was detected for ethanol sensitivity (Figs 4D'-E', S4B'). Hence, the APL neurons function specifically in rapid tolerance development. By contrast, GABAergic signaling in the DPM neurons and gap junctions between the APL and DPM neurons promote ethanol sensitivity (S4D'-F' Fig).

## AST requires GABAergic repression of Kenyon cells

Because APL activity supports acquisition of rapid tolerance and APL neurons are GABAergic, we tested for the role of GABA receptivity in the mushroom body Kenyon cells. *Drosophila* Kenyon cells express ionotropic GABA$_A$ receptors and metabotropic GABA$_B$ receptors, and both receptors function in ethanol tolerance [55]. Moreover, the *Resistance to dieldrin* (*Rdl*) subunit of GABA$_A$ regulates acquisition of both aversive and appetitive memories in Kenyon cells [56–58]. RNAi against *Rdl* in all Kenyon cells decreased rapid tolerance measured at 4 hr, but not at 24 hr (Fig 5A). Thus, activity of the GABAergic APL neurons but not the GABAergic DPM neurons (S4D, E Fig) promotes acquisition of rapid tolerance, and mushroom body Kenyon cells receive GABAergic input to promote AST.

Given the multiple ways that mushroom body circuitry regulates rapid tolerance, we hypothesized that Kenyon cell neural activity is also required. Surprisingly, acute inactivation of synaptic release of all Kenyon cells did not affect rapid tolerance (Fig 5B). Simultaneous inactivation of all Kenyon cells might mask roles for the individual lobes. However, separate

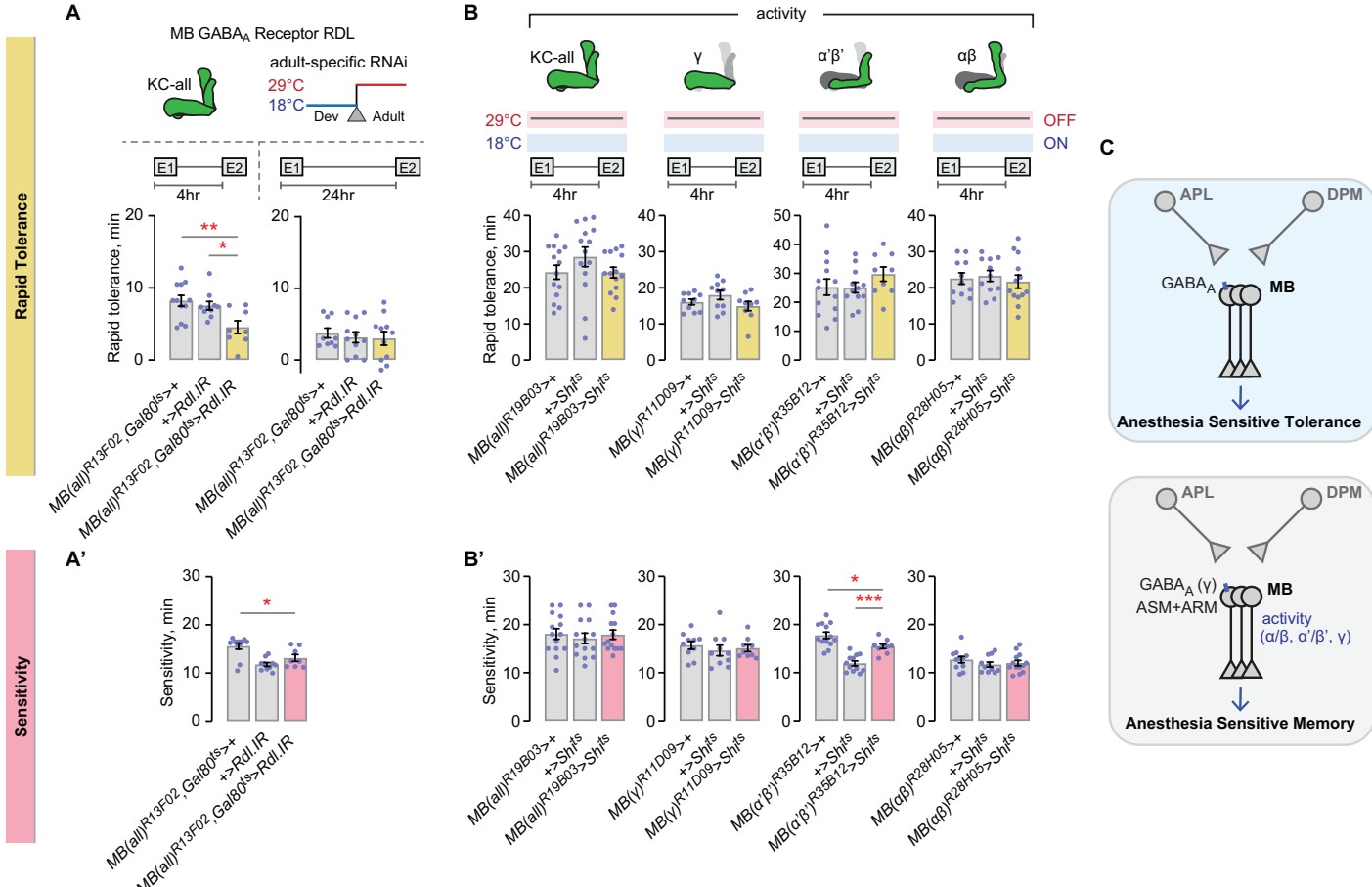

**Fig 5. AST requires GABAergic repression of the Kenyon cells, but not ongoing neuronal activity. A, A'**) GABA$_A$ receptor subunit *Rdl* is required to promote AST in mushroom body Kenyon cells, with no effect on sedation sensitivity. Tolerance, left: one-way ANOVA/Holm-Šídák's; right: one-way ANOVA. Sensitivity: one-way ANOVA/Holm-Šídák's. **B, B'**) Neuronal activity is not required in the mushroom body Kenyon cells for rapid tolerance or sedation sensitivity. Tolerance, left: Kruskal-Wallis; center-left: one-way ANOVA; center-right: one-way ANOVA; right: one-way ANOVA. Sensitivity, left: Kruskal-Wallis; center-left: one-way ANOVA; center-right: one-way ANOVA/Holm-Šídák's; right: one-way ANOVA. **C**) Role of GABAergic signaling and neuronal activity in the mushroom body Kenyon cells for AST and ASM.

inactivation of the γ, α'β', and the αβ lobes did not affect rapid tolerance (Fig 5B). Thus, synaptic release from Kenyon cells is dispensable for rapid tolerance.

To conclude, adult Kenyon cells require GABA$_A$ receptors but not neuronal activity to support AST, suggesting that rapid tolerance occurs via GABAergic inhibition of Kenyon cells. By contrast, classical ASM and ARM are well-mapped to Kenyon cell lobes, for example ongoing neuronal activity of γ neurons supports ASM and not ARM (Fig 5C) [41].

## Kenyon cells are dispensable for protein synthesis-dependent chronic tolerance

We next compared chronic tolerance and long-term memory mechanisms in mushroom body circuitry. Mushroom body circuitry is critical for the acquisition, consolidation, and expression of LTM, and aspects of the circuitry are critical for alcohol associative preference arising from LTM-like spaced training [9]. LTM of associatively-trained events requires new protein synthesis, as does chronic ethanol tolerance [11,45,59–61]. The site of action for new protein synthesis is not known for chronic tolerance. We followed up on translation-dependent chronic tolerance using a genetically encoded eukaryotic ribosome inhibitor, the Ricin toxin, that was made cold-sensitive to allow for spatiotemporal inducibility [62–65]. Neuronal Ricin activation is functionally equivalent to cycloheximide in blocking LTM [62]. We first expressed Ricin in all adult neurons and yielded testable flies when combined with *Gal80$^{ts}$* that blocks unwanted transgene expression at the restrictive temperature (18°C). Adult-specific, pan-neuronal inhibition of protein translation caused decreased chronic tolerance (Fig 6A), confirming that chronic tolerance is protein synthesis-dependent and localizing it to the nervous system. Blocking new protein synthesis during the entire chronic tolerance paradigm in all adult Kenyon cells did not affect chronic tolerance (Fig 6B). Similarly, blocking adult Kenyon cell protein translation specifically during the tolerance-inducing dose of chronic ethanol, or during the subsequent 24 hr inter-exposure interval, caused no changes in chronic tolerance (Fig 6B). Thus, a protein synthesis-dependent trace for chronic tolerance exists in a non-Kenyon cell site in the adult *Drosophila* nervous system.

CREB, that is critical for classical LTM in specific Kenyon cell lobes, functions outside the mushroom bodies for chronic tolerance, mirroring the localization for new protein translation in chronic tolerance [5,10,63]. To determine if the transcriptional and post-translational pathways participating in CREB encoding are also dispensable in Kenyon cells, we tested the role of Mef2, a transcription factor that induces CREB expression, and also Ca$^{2+}$ signaling-associated genes, in our chronic tolerance assay [66]. Mef2 signaling is critical for rapid tolerance in the αβ Kenyon cells via induction of the immediate early gene (IEG) *Hr38* [67]. Expressing dominant negative Mef2 in all Kenyon cells, however, did not affect chronic tolerance or sensitivity, thus ruling out Mef2-mediated signaling, and further distinguishing rapid and chronic ethanol tolerance (Fig 6C). The phosphorylation of CREB protein is a consequence of Ca$^{2+}$ influx, cAMP generation, and activation of the MAPK signaling cascade. Intracellular calcium levels are regulated in part by entrance through NMDA receptors, thus knockdown of the constituent subunit *Nmdar1* abolishes NMDA-mediated synaptic plasticity. Kenyon cell-specific knockdown of *Nmdar1* did not affect chronic tolerance, ruling out one possible CREB activation route (Fig 6C). Interestingly, aversive long-term memory also does not require NMDARs in the mushroom bodies; instead, they act in ellipsoid body neurons [45]. To test Ca$^{2+}$-sensitive kinase activation in Kenyon cells, we expressed a constitutively active version of CaMKII, *CaMKII$^{T287D}$* [68]. However, no change in chronic tolerance was observed, suggesting that a canonical calcium-dependent kinase response is dispensable in Kenyon cells (Fig 6C). Last, we asked if any cAMP generation is important for chronic tolerance using a mutant for the *Drosophila* adenylyl cyclase *rut* gene. *Rut$^1$* mutants exhibit

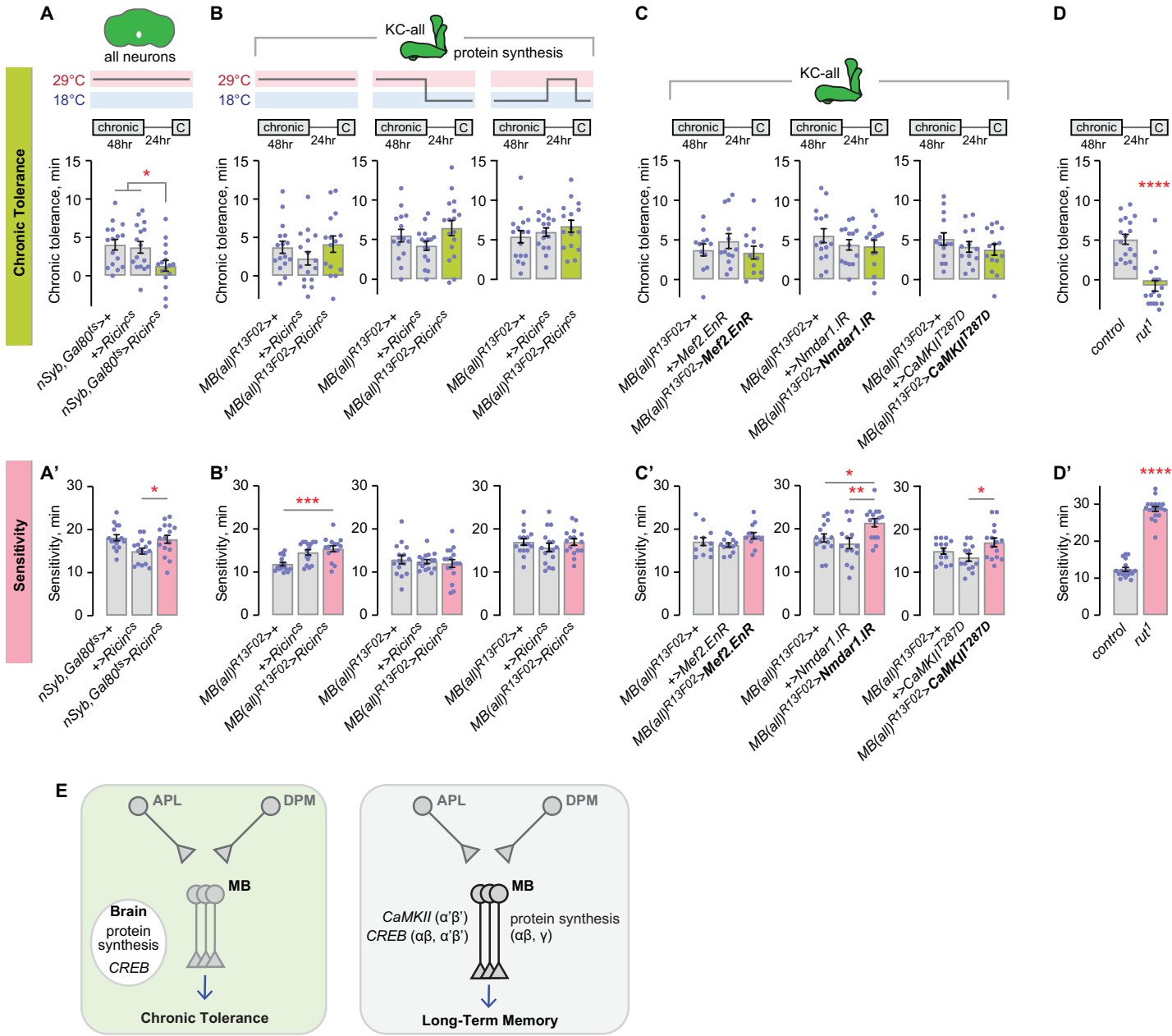

**Fig 6. Chronic tolerance requires protein synthesis in adult neurons but not in the mushroom body Kenyon cells. A, A'**) Protein synthesis in neurons is required during the chronic tolerance paradigm. No effect on sedation sensitivity. Tolerance: one-way ANOVA/Holm-Šídák's. Sensitivity: one-way ANOVA/Holm-Šídák's. **B, B'**) Protein synthesis is not required in the mushroom body Kenyon cells for chronic tolerance or for sedation sensitivity. Tolerance, left and center: one-way ANOVA; right Brown-Forsythe. Sensitivity, left: One-way ANOVA/Holm-Šídák's; center: Brown-Forsythe; right: one-way ANOVA. **C, C'**) Left: dominant negative *Mef2* in the mushroom body Kenyon cells does not affect chronic tolerance or sedation sensitivity. One-way ANOVA. Middle: *Nmdar1* is not required in the mushroom body Kenyon cells for chronic tolerance. It promotes sedation sensitivity there. Tolerance: one-way ANOVA. Sensitivity: one-way ANOVA/Holm-Šídák's. Right: constitutively active *CaMKII* in the mushroom body Kenyon cells does not affect chronic tolerance or sedation sensitivity. Tolerance: one-way ANOVA. Sensitivity: one-way ANOVA/Holm-Šídák's. **D, D'**) The *rut* adenylyl cyclase is required for chronic tolerance and sedation sensitivity. Tolerance: Mann-Whitney test (two-tailed). Sensitivity: unpaired t test (two-tailed). **E**) Chronic tolerance molecular encoding mechanisms occur outside the mushroom body circuitry, in contrast to classical LTM.

strongly reduced chronic tolerance and sensitivity, consistent with its role in supporting associative learning (Fig 6D, D').

Taken together, chronic tolerance is CREB- and protein synthesis-dependent but it differs from classical LTM in that these functions reside outside the mushroom bodies. (Fig 6E).

## DPM activity during chronic ethanol exposure induces chronic tolerance

To continue mapping the genetic and circuit features of chronic ethanol tolerance, we searched for roles for other neuronal constituents of the mushroom body circuitry [10,11]. The DPM neurons consolidate classical LTM [69–71]. Acute inactivation of the DPM neurons throughout the chronic tolerance paradigm reduced chronic tolerance (Fig 7A). Limiting DPM blockade to the inter-exposure interval, however, had no effect on chronic tolerance (Fig 7A). To more completely ask when the DPMs temporally regulate chronic tolerance at all, we used the more specific *DPM$^{VT}$-Gal4* driver and tested each of the three phases. Inactivation of the DPMs during the chronic ethanol exposure, but not the inter-exposure interval or during expression, caused decreased chronic tolerance (Fig 7B). Thus, neuronal activity of the DPMs during the chronic ethanol exposure promotes tolerance development.

Blocking neural activity in the APL neurons or in all Kenyon cells during the entire chronic paradigm did not affect chronic tolerance (Fig 7C, D). Moreover, separately blocking activity in each mushroom body lobe resulted in no change in chronic tolerance, although there was a trend towards increased tolerance when the γ lobes were inactivated ($P$=0.0697, experimental *vs. Gal4* control, $P$=0.0476, experimental *vs. UAS* control) (Fig 7D). Hence, chronic tolerance appears to not require neuronal activity in the mushroom body circuitry, in contrast to classical long-term memories (Fig 7E).

Two manipulations affected ethanol sensitivity: inactivation of the DPM neurons for 48 hr one day prior to the ethanol challenge dose, and 72 hr inactivation of the αβ Kenyon cells (Fig 4B', D'). These data suggest that initial ethanol sensitivity may require activity in specific brain networks prior to ethanol exposure.

## DPM GABAergic repression of Kenyon cells is compartmentalized to support chronic tolerance

DPM neurons are GABAergic and serotonergic, and they release the Amn neuropeptide [35,38,72]. Aversive spaced training LTM requires the GABA$_A$ receptor Rdl in the mushroom body Kenyon cells [47]. RNAi against the Gad1 GABA synthetic enzyme in the DPMs decreased chronic tolerance (Fig 8A). Water-reward LTM requires serotonin synthesis in the DPMs [73]. Adult-specific RNAi against the serotonin synthesis gene *Ddc* in the DPMs did not affect chronic tolerance (S5A, A' Fig). Amn is dispensable for chronic tolerance [10]. Thus, it is likely that the DPMs regulate chronic tolerance specifically through GABA release.

To map possible targets of DPM GABA signaling, we tested the role of the GABA$_A$ receptors. Adult-specific RNAi against the *Rdl* GABA$_A$ subunit in all Kenyon cells reduced chronic tolerance (Fig 8B). Expressing *Rdl* RNAi in αβ Kenyon cells caused decreased chronic tolerance (Figs 8C, S5B). No difference in chronic tolerance was observed when *Rdl* RNAi was expressed in other Kenyon cell types, or in the APL or DPM neurons (Figs 8C, S5C). Thus, GABAergic release from the DPMs during chronic ethanol exposure likely compartmentalizes to the αβ Kenyon cells via ionotropic GABA$_A$ receptors (Fig 8D).

Ethanol sensitivity required GABA synthesis in the DPM neurons and the Rdl GABA$_A$ receptor in αβ Kenyon cells (Fig 8A', C'). However, GABA$_A$ receptors appeared to be required during development for ethanol sensitivity, in contrast to its adult role for chronic tolerance (Fig 8B').

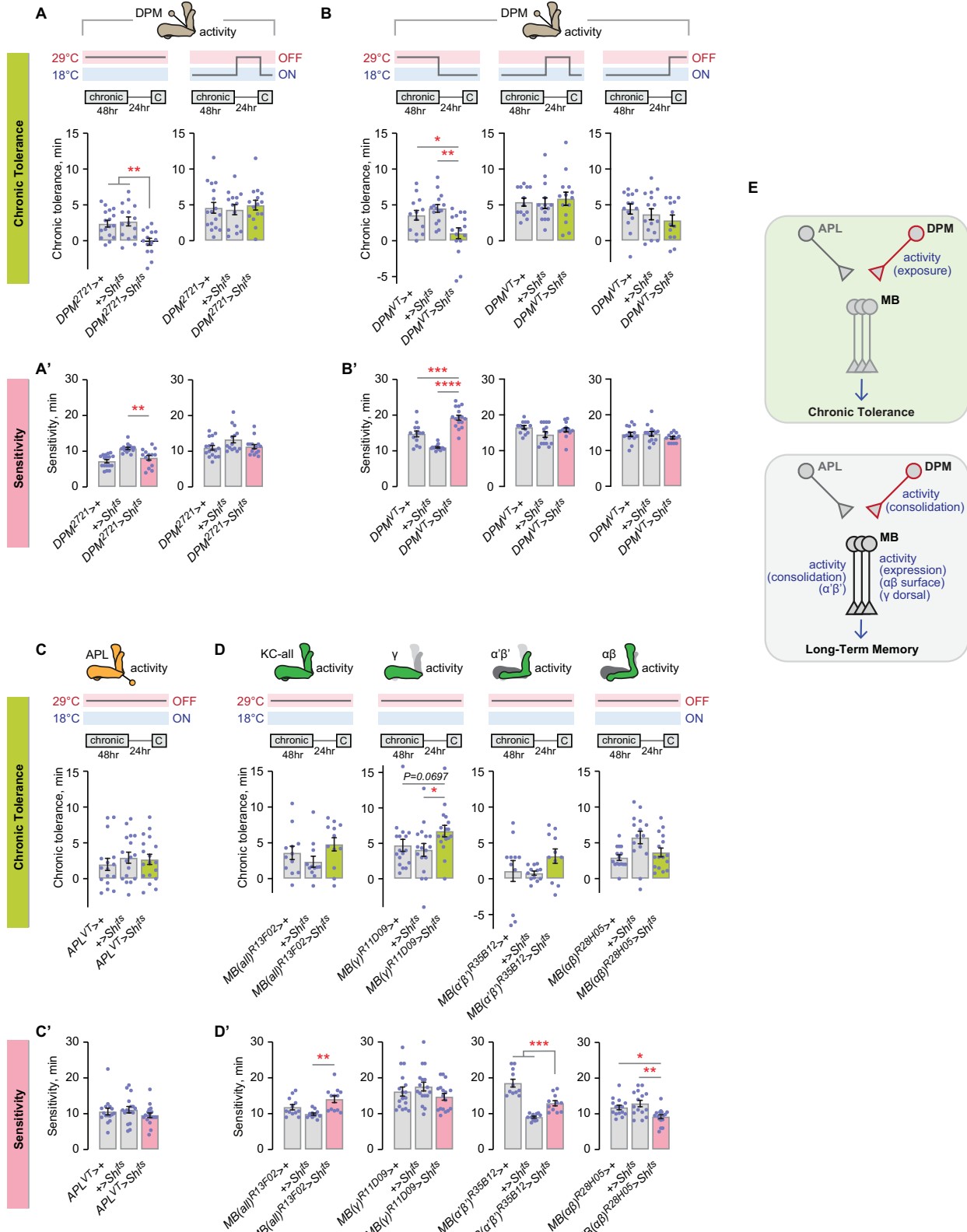

**Fig 7. DPM synaptic release is required during chronic ethanol exposure for chronic tolerance. A, A'**) DPM synaptic release is required during the chronic paradigm for chronic tolerance development, and it is not required for sedation sensitivity. Tolerance, left: one-way ANOVA/Holm-Šídák's; right: one-way ANOVA. Sensitivity, left: Brown-Forsythe/Dunnett's; right: Kruskal-Wallis. **B, B'**) DPM synaptic release

is specifically required during chronic ethanol exposure for tolerance development. Tolerance, left: one-way ANOVA/Holm-Šídák's; center and right: one-way ANOVA. Sensitivity, left: Brown-Forsythe/Dunnett's; center: Brown-Forsythe; right: one-way ANOVA. Sensitivity, left: Brown-Forsythe/Dunnett's; center: Brown-Forsythe; right: one-way ANOVA. **C, C'**) APL synaptic release is dispensable for chronic tolerance and for sedation sensitivity. Tolerance: one-way ANOVA. Sensitivity: Kruskal-Wallis. **D, D'**) Mushroom body Kenyon cell synaptic release, tested in each lobe, is dispensable for chronic tolerance. Synaptic release in the αβ Kenyon cells is required for ethanol sensitivity. Tolerance, left: one-way ANOVA; center-left: Kruskal-Wallis/Dunn's; center-right: Brown-Forsythe; right: Brown-Forsythe/Dunnett's. Sensitivity left: Brown-Forsythe/Dunnett's; center-left: Kruskal-Wallis; center-right: Brown-Forsythe/Dunnett's; right: one-way ANOVA/Holm-Šídák's. **E**) Role of synaptic release in the mushroom body circuitry for chronic tolerance and long-term memory.

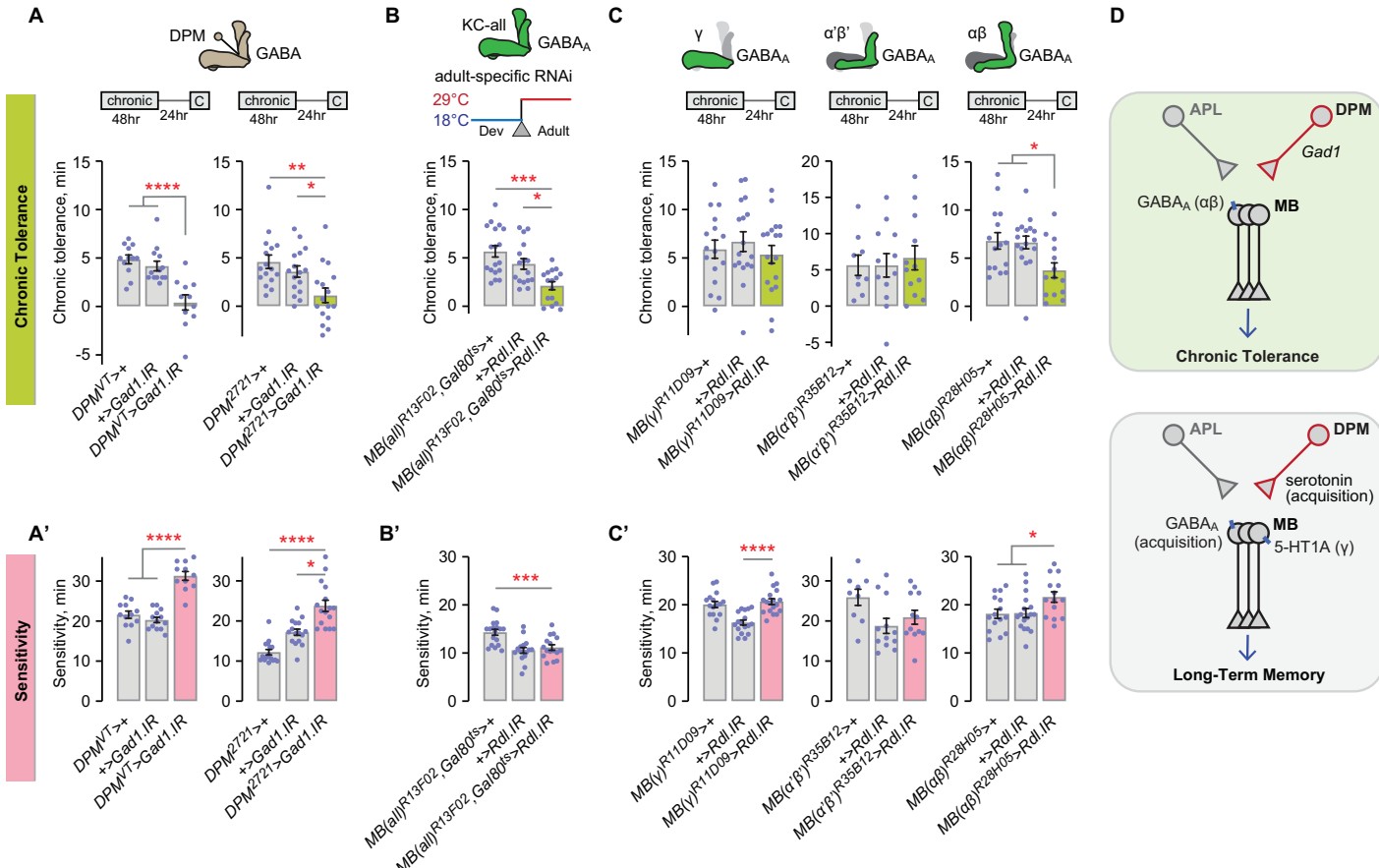

**Fig 8. DPM GABAergic repression of mushroom body Kenyon cells is compartmentalized to the αβ lobes to promote chronic tolerance. A, A'**) GABA synthesis is required in the DPM neurons to promote chronic tolerance and sedation sensitivity. Tolerance, left: one-way ANOVA/Holm-Šídák's; right: Kruskal-Wallis/Dunn's. Sensitivity, left: one-way ANOVA/Holm-Šídák's; right: Kruskal-Wallis/Dunn's. **B, B'**) *Rdl* in the adult mushroom body Kenyon cells promotes chronic tolerance, but not sedation sensitivity. Tolerance: Kruskal-Wallis/Dunn's. Sensitivity: one-way ANOVA. **C, C'**) *Rdl* in the αβ Kenyon cells promotes chronic tolerance and sedation sensitivity. Tolerance, left and center: one-way ANOVA; right: one-way ANOVA/Šídák's. Sensitivity, left, center, and right: one-way ANOVA/Šídák's. **D**) Summary comparison of DPM roles in chronic tolerance and long-term memory.

## DPMs require protein synthesis, not CREB, to support chronic tolerance

Because CREB functions outside the Kenyon cells for chronic tolerance, we tested if it acts in the DPM neurons [10]. Dominant negative CREB expressed in the DPMs did not affect chronic tolerance (Fig 9A). Blocking protein synthesis in the DPMs during the chronic ethanol exposure, but not the inter-exposure interval, reduced chronic tolerance (Fig 9A). Thus, a protein synthesis-dependent but CREB-independent chronic tolerance trace exists

in the DPMs. Since the αβ Kenyon cells are the likely target of DPM GABAergic signaling, we tested whether they too require protein synthesis. Protein synthesis in αβ neurons was dispensable for chronic tolerance during the chronic exposure and during the inter-exposure interval (Fig 9B). Finally, we investigated the role of the IEG *kay*—homolog of mammalian

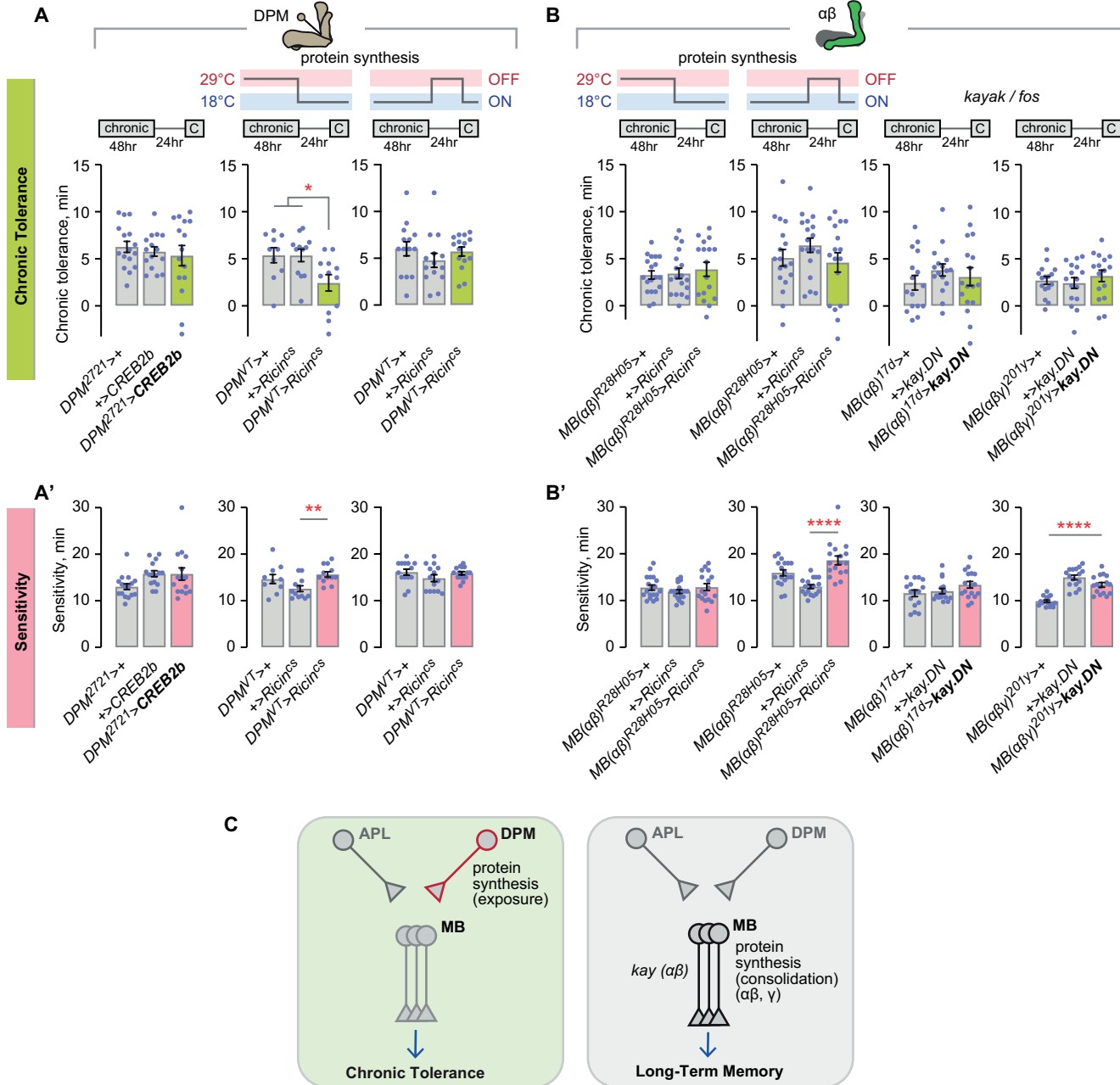

**Fig 9. DPM neurons require protein synthesis, but not CREB, to promote chronic tolerance. A, A'**) DPM neurons require protein synthesis during the chronic exposure, but they do not depend on CREB for chronic tolerance. Sedation sensitivity is unaffected. Tolerance, left: Brown-Forsythe; middle: Kruskal-Wallis/Dunn's; right: one-way ANOVA. Sensitivity, left: Kruskal-Wallis; middle: one-way ANOVA/Šídák's; right: Brown-Forsythe. **B, B'**) Neither protein synthesis nor the *kayak* immediate early gene are required in the mushroom body αβ Kenyon cells for chronic tolerance or sedation sensitivity. Tolerance, left: Brown-Forsythe; all others: one-way ANOVA. Sensitivity, left: Brown-Forsythe; center-left: Kruskal-Wallis/Dunn's; center-right: one-way ANOVA; right: Brown-Forsythe/Dunnett's. **C**) Summary diagram of protein synthesis dependence in the mushroom bodies for chronic tolerance and LTM.

*c-fos*—that marks LTM engram neurons and is responsible for the persistent CREB expression that sustains LTM in αβ Kenyon cells [74]. *kay* is selectively induced by an inebriating ethanol exposure following chronic ethanol exposure [10]. Dominant negative *kay* expressed in αβ Kenyon cells did not affect chronic tolerance (Fig 9B). There was no effect of any manipulation on ethanol sensitivity (Fig 9A', B'). Thus, chronic ethanol tolerance requires protein synthesis and GABA release from the DPM neurons during the chronic ethanol exposure and GABA reception by αβ Kenyon cells (Fig 9C).

## Sirt1 suppresses formation of a labile ethanol memory that consolidates into enhanced chronic tolerance

The histone/protein deacetylase Sirt1 in the γ lobe Kenyon cells inhibits excess chronic tolerance [10]. We sought to determine the role of Sirt1 with respect to tolerance memory-like states. We first determined the type of ethanol memory-like state that Sirt1 suppresses. In wild-type controls, chronic tolerance is comprised of an LTM-like state, but not AST or ART, as evidenced by cold shock anesthesia insensitivity, and no effect of mutation of either *amn* or *rsh* [10]. Flies lacking *Sirt1* and flies expressing RNAi against *Sirt1* in γ lobe Kenyon cells both exhibited enhanced chronic tolerance measured at 24 hr (Fig 10A, and as previously reported) [10]. Cold shock anesthesia delivered 30 min after cessation of the chronic ethanol exposure reverted the enhanced tolerance of *Sirt1* mutants to control levels, indicating that *Sirt1* normally suppresses an anesthesia-sensitive trace created by chronic ethanol exposure (Fig 10A). However, cold shock was ineffective in suppressing *Sirt1* enhanced tolerance when delivered 30 min prior to the challenge dose (Fig 10A). RNAi against *Sirt1* in the γ lobe Kenyon cells was also sensitive to cold shock anesthesia delivered 30 min after cessation of the chronic ethanol exposure, localizing the *Sirt1*-suppressable anesthesia sensitive trace to the γ neurons (Fig 10B). We asked if this Sirt1-dependent enhanced tolerance persists beyond the 72 hr decay time previously observed in wild-type control flies [10]. At that late timepoint, neither *Sirt1* null mutants nor flies with *Sirt1* RNAi in γ Kenyon cells showed increased tolerance (Fig 10C, D). Finally, in addition to inhibiting chronic tolerance, *Sirt1* also promotes rapid tolerance in the αβ Kenyon cells [30]. However, the role of *Sirt1* in other Kenyon cells is not known for rapid tolerance. RNAi against *Sirt1* in the γ lobe Kenyon cells did not affect rapid tolerance, nor did a cold shock unmask a role (Fig 10E). As observed previously, *Sirt1* loss of function reduced sedation sensitivity (Fig 10A) [75]. Thus, the functions of *Sirt1* in chronic and rapid ethanol tolerance are distinct. *Sirt1* normally blocks the formation of a labile trace in the γ Kenyon cells, and this trace consolidates within 24 hr, enhancing chronic tolerance (Fig 10F).

## Discussion

A comparative summary of our findings is presented in Fig 11. The main conclusions from our experiments are 1) a near complete separation of the neural circuitry for three ethanol tolerance memory-like states, 2) delineation of the role of mushroom body learning and memory center pathways for ethanol tolerance and sensitivity, 3) considerable separation of the mechanisms for ethanol tolerance memory-like states and classical aversive learning and memory circuit mechanisms, and 4) identification of a potential mechanism to regulate how chronic ethanol is encoded into a lasting experience. Other mushroom body extrinsic neurons, including dopamine neurons and mushroom body output neurons, may contribute to ethanol tolerance memory-like state circuits.

For rapid tolerance, a sedating dose of ethanol induces two time-dependent components of tolerance that are distributed across the *Drosophila* brain. First, an AST trace occurs in Kenyon cells, as it requires *Rdl* and *Nmdar1* in all lobes, and *amn* in just the γ lobe. One way

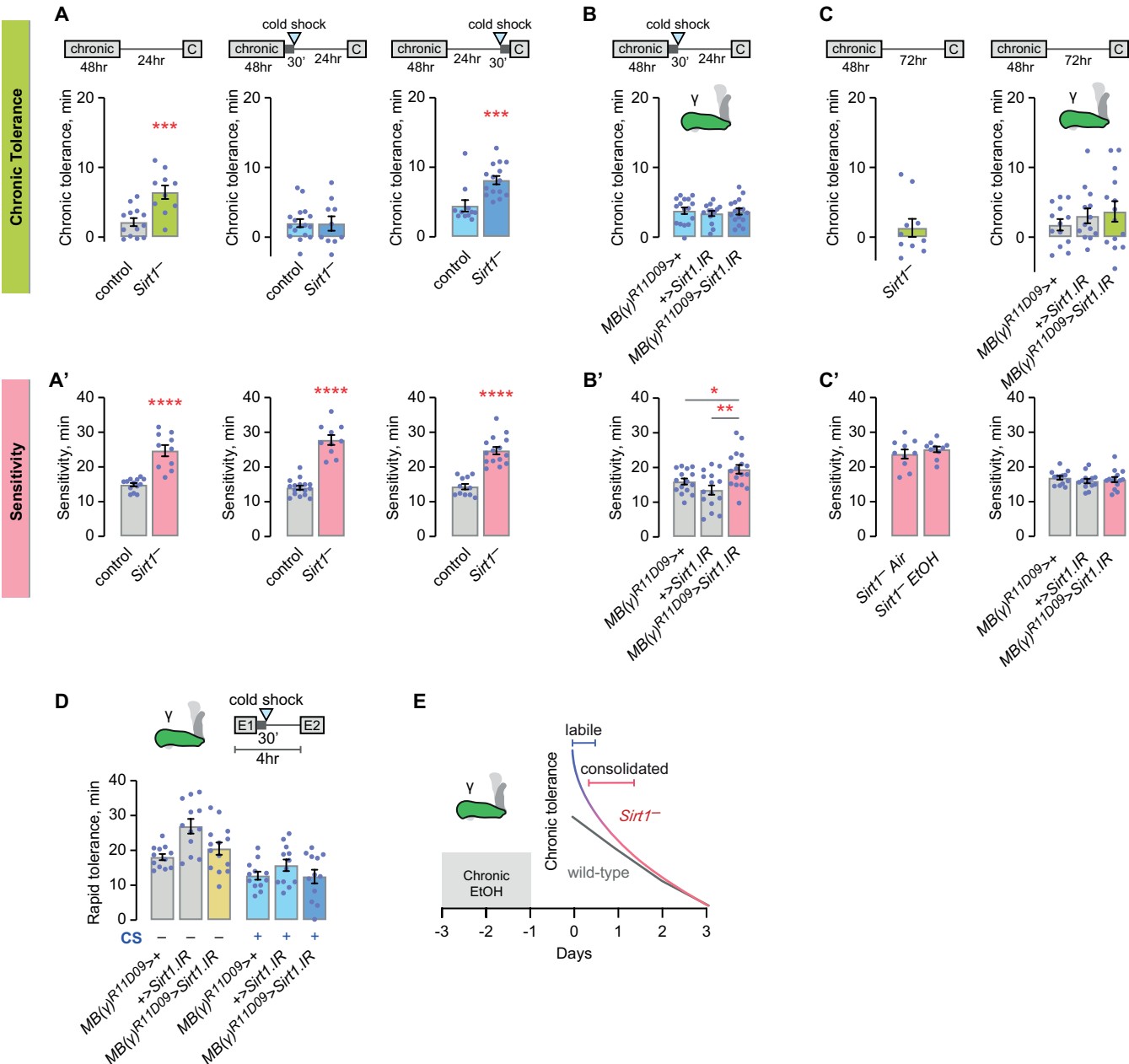

**Fig 10. Sirt1 suppresses formation of a labile trace in the mushroom body γ lobe Kenyon cells that consolidates into enhanced chronic tolerance. A, A')** *Sirt1* null mutants exhibit higher chronic tolerance that is anesthesia sensitive early but not late during the inter-exposure interval. Tolerance, left: unpaired t test (two-tailed); middle: unpaired t test (two-tailed); right: Mann-Whitney test (two-tailed). Sensitivity, left: Mann-Whitney test (two-tailed); middle: Mann-Whitney test (two-tailed); right: unpaired t test (two-tailed). **B, B')** Labile chronic tolerance trace localizes to the mushroom body γ lobes. Tolerance: one-way ANOVA. Sensitivity: one-way ANOVA/Holm-Šídák's. **C, C')** *Sirt1* null mutant enhanced chronic tolerance is dissipated by 72 hr. Tolerance, left: one-sample t test, compared to zero; right: one-way ANOVA. Sensitivity, left: unpaired t test (two-tailed); right: one-way ANOVA. **D)** No *Sirt1* null-dependent enhancement of rapid tolerance in the mushroom body γ lobes. Left: Brown Forsythe; right: one-way ANOVA. **E)** Model for Sirt1-suppression of a labile trace that consolidates to enhance chronic tolerance.

that AST differs from classical ASM is the latter requires *amn* in αβ Kenyon cells [43]. Second, an ART trace occurs via both APL induction and *rsh* in a non-mushroom body site. Ethanol ART is distinct from aversively conditioned ARM because *rsh* acts in Kenyon cells for ARM but not ART.

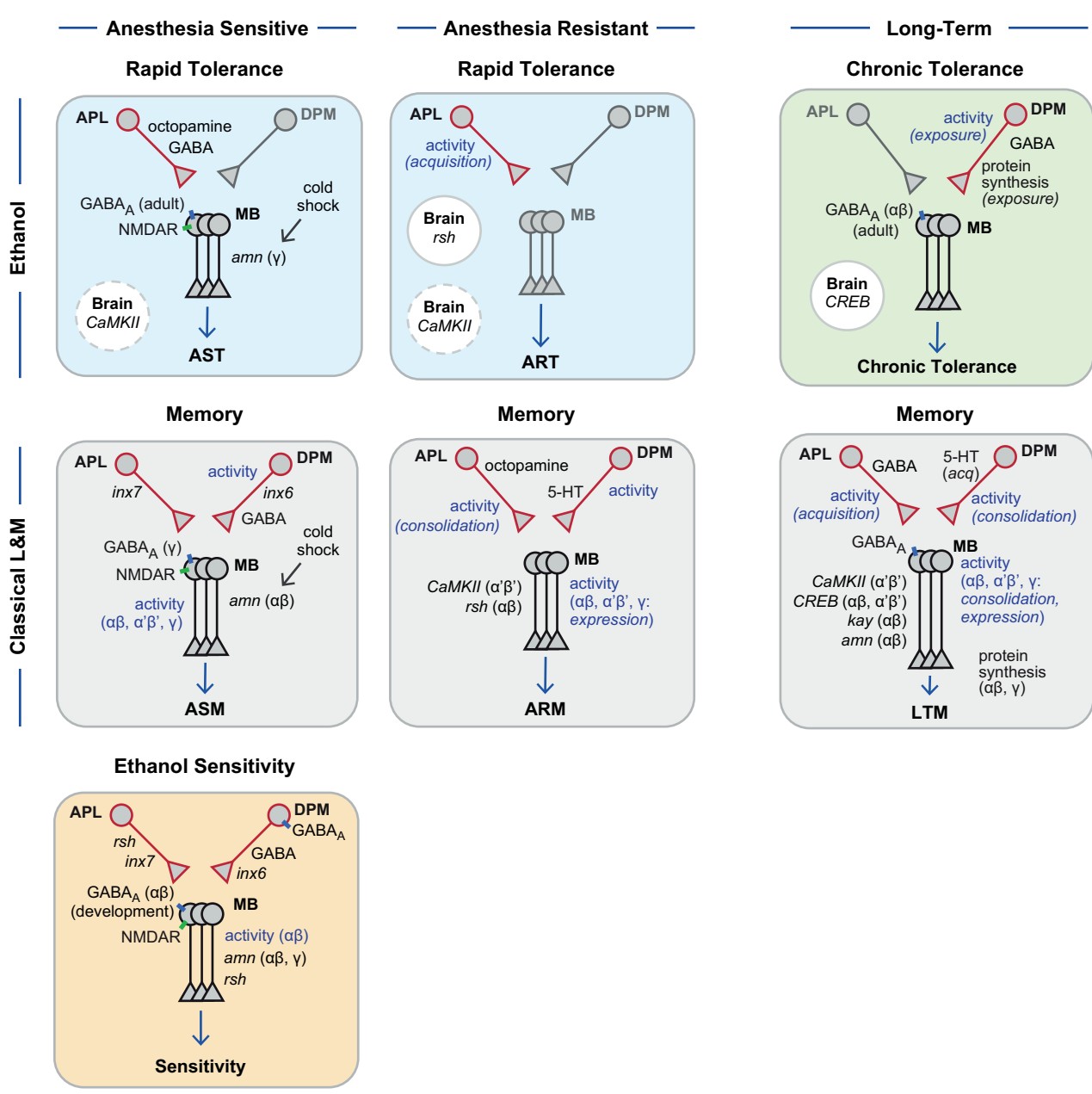

**Fig 11. Summary diagrams of mushroom body genes and circuits for *Drosophila* ethanol behaviors and classical forms of associative memory.**
Blue (rapid) and green (chronic) boxes contain pathways for ethanol tolerance and the orange box for ethanol sensitivity. Gray boxes contain pathways for associative memory that were tested for ethanol behavior.

For chronic tolerance, a continuous low dose of ethanol engages a separate inhibitory circuit in the *Drosophila* mushroom bodies. The DPMs release GABA, likely inhibiting αβ Kenyon cells during the chronic ethanol exposure, and this silencing may prevent their activity, protein synthesis encoding, and *kay*-encoding capacity. This is a major difference between chronic tolerance and classical LTM: αβ Kenyon cells classically hold aversive and appetitive LTM [63,71,74,76]. Consolidation of chronic tolerance may occur during the 48 hour continuous low dose ethanol exposure and not during the inter-exposure interval. A requirement for

protein synthesis in the DPM neurons during the chronic ethanol exposure but not during the inter-exposure interval supports this notion.

Our findings indicate that the Kenyon cells play a less central role than previously thought for tolerance formation and storage. For example, prior evidence showed that hyperpolarization of all Kenyon cells with the Kir2 inwardly rectifying potassium channel abolishes chronic tolerance, whereas here blocking synaptic release in the same population had no effect [10]. Many factors may account for the apparent discrepancy. For example, hyperpolarization may affect the voltage-sensitive GABA$_A$ receptors that are required for chronic tolerance. Alternatetively, Kir2 hyperpolarization was continuous during adulthood, whereas synaptic release was blocked just before ethanol administration. For rapid tolerance, neural synaptobrevin that is essential for synaptic release is required in the Kenyon cells, as is the Amn neuropeptide, but again here we found that synaptic release from the Kenyon cells was dispensable during the rapid tolerance paradigm [30,77]. Further research, development of the effector tools that probe function, and refinement of the neuronal drivers that target key brain regions, will make clearer the genetic and circuit functions of intrinsic and extrinsic mushroom body neurons on ethanol behaviors.

The AST-specific role of *amn*, *rut*, and *Nmdar1* is of particular interest. Indeed, many cAMP signaling components display deficits both in learning and memory and in ethanol behaviors when mutated, such as *rut*, *PKA*, and *elm* [11,78,79]. Dysregulated cAMP signaling alters both consolidated and labile forms of classical memory. Protein kinase A (PKA) in the mushroom bodies specifically inhibits ARM, and the *dunce* phosphodiesterase, that degrades cAMP, acts at multiple nodes in the olfactory pathway to support ARM, not ASM [80,81]. Conversely, PKA anchoring proteins and the Rut adenylyl cyclase function in the mushroom bodies to support ASM, not ARM [81,82]. Uncovering how these cAMP pathway components are activated or inhibited by the length and dose of ethanol intake will inform and help predict what specific downstream changes are important for tolerance.

Both ART and chronic tolerance exemplify consolidated tolerance because they are unaffected by amnestic cold shock. Does chronic tolerance have any ARM components to it? Our chronic ethanol paradigm appears phenomenologically like a massed training protocol that creates ARM. However, current evidence suggests that chronic tolerance is distinct from ARM in key ways. First, there are two types of ARM: one that is produced by single cycle training and is not sensitive to cold shock, while the other is produced by massed training and is sensitive to cold shock [83]. Both ARM types do not require protein synthesis. Both ARM types do require the *rsh* GTPase [40]. In contrast, chronic ethanol creates tolerance that is resistant to amnestic cold shock at 3 hr and 24 hr timepoints [10]. Also, chronic tolerance does require protein synthesis, and it does not require *rsh* [10,11]. Therefore, chronic tolerance shares some temporal properties, but lacks the mechanistic components, of ARM. Adopting tests that erase labile memories into the alcohol behavior field has revealed a possible mode of intervention for alcohol use disorder. For example, psychiatric procedures that reduce the intensity of persistent fear memories employ reconsolidation-mediated lability on these inappropriately learned events [84]. Forcing tolerance states to become labile to intentionally erase them may rescue ethanol sensitivity and ultimately reduce further drinking.

Inhibitory circuits in the mushroom bodies are a site for ethanol's pharmacological actions. Inhibitory circuits are also critical for olfactory learning, partly because APL inhibition contributes to odor discrimination by sparsening odor representation in the Kenyon cells [48,49,85]. Even if the APL neurons promote sparse encoding of ethanol as an olfactory stimulus, it is unlikely that this olfactory discrimination is responsible for tolerance. First, the APL neurons are dispensable for chronic tolerance. Second, APL GABA signaling is dispensable for ART. Third, cross tolerance to the sedating effects of alcohol exists between ethanol and other

odorants with sedative properties, such as benzyl alcohol, suggesting pharmacological action [86]. The ability of ethanol to diffuse through membranes and affect all cells likely presents a unique challenge for the brain to encode the drug experience. For chronic tolerance, the specific encoding into the DPM neurons may be due to both the volatility of ethanol and the long duration of the exposure paradigm [52].

The APL (rapid tolerance) and DPM (chronic tolerance) neurons likely promote tolerance through inhibition. Silenced Kenyon cells may prevent a coincidence detection mechanism from occurring, that would otherwise drive associative learning of internal and/or environmental cues associated with ethanol inebriation. Indeed, locating a calcium response in the mushroom bodies due to acute ethanol exposure has proven challenging [87]. Ethanol can cause behavioral plasticity via Kenyon cells and extrinsic mushroom body circuitry: associative preference or aversion arising from spaced training with inebriation requires sequential mushroom body circuits [7,9]. Moreover, shutdown of the mushroom bodies may be important for tolerance encoding by favoring a separate circuit in the *Drosophila* brain. For example, the DPMs promote sleep by inhibiting wake-promoting α'β' neurons with GABA [72]. A long-term goal will be to understand interactions between apparently disparate behavioral states that use overlapping circuits, such as DPM-Kenyon cell function in sleep and chronic tolerance.

Sirt1 provides a new and different type of regulated memory inhibition mechanism in the mushroom body circuitry. Sirt1 expression levels are readily changed by drugs of abuse, including ethanol [75,88]. Chromatin regulators and transcriptional machinery shift over the course of initial LTM storage and long-term maintenance [89,90]. We found that Sirt1 specifically and only in the γ Kenyon cells inhibits the creation of an extra chronic tolerance memory-like state. It is possible that other Sirt1-associated behaviors, including circadian rhythms and depressive-like states, may regulate the spectrum of memory-like states that ethanol engages via regulation of Sirt1, thus altering the encoding of ethanol experience [91,92].

Our findings reveal relationships between ethanol sensitivity and the development of two of the three different forms of tolerance, AST and chronic. It is well documented that longer exposure and higher concentration ethanol lead to coupled increases in ethanol sensitivity and rapid tolerance in *Drosophila* [10,11,13,75]. Studies of genetic mutants may uncover a similar correlation ([93], but see [94]). There are clues that ethanol sensitivity and tolerance can be separated in some cases with detailed understanding of specific genes and neural substrates, for example by determining the temporal requirement (developmental vs. adult) or by refinement of the site of action of a gene [10,77]. Our findings support the existence of both singular and multifaceted roles for individual genes in ethanol responses. This may have useful implications for the understanding of the genetic basis for alcohol use disorder in humans, and for its eventual treatment: genes that impact both sensitivity and tolerance may be targeted to regulate multiple alcohol responses. Moreover, much more efficient screens can be done for ethanol sensitivity in organisms like *Drosophila* that model endophenotypes of alcohol use disorder, to enrich for genes that also impact ethanol induced plasticity.

Ethanol sensitivity and AST partially overlap, whereas ethanol sensitivity and ART showed no overlap. Sensitivity and AST require NMDAR1 in the Kenyon cells, and the Amn neuropeptide in the γ lobe Kenyon cells. Thus, ethanol sensitivity appears to be related to the temporally shorter and labile ethanol memory-like state. AST, but not sensitivity, requires octopamine and GABA in the APL neurons, and likely requires CaMKII in neurons. An important further difference is revealed by temporal mapping: the Rdl GABA$_A$ receptor is required during adulthood for AST but during development for ethanol sensitivity in the Kenyon cells. Thus, there may exist sensitivity-dependent and -independent AST pathways in the mushroom body circuitry. Additional experiments may further separate ethanol sensitivity and AST.

Ethanol sensitivity and chronic tolerance functional overlap is limited to GABA release from the DPM neurons and GABA$_A$ reception in the αβ lobe Kenyon cells. This same GABAergic DPM-to-Kenyon cell putative pathway is dispensable for AST and ART, arguing against a derepressed representation of ethanol's effects in the Kenyon cells causing defects in tolerance development.

Ethanol sensitivity maps more extensively in the mushroom body circuitry than any form of tolerance. It requires gap junction communication between the APL and DPM neurons, the Rdl GABA$_A$ receptor in the DPM neurons, the Rsh GTPase in the APL neurons and the Kenyon cells, and synaptic release and the Amn neuropeptide in the αβ Kenyon cells. We suspect that ethanol sedation sensitivity, an innate physiological response, is influenced by more factors than the acquired state of ethanol tolerance. Interestingly, ethanol sensitivity and classical anesthesia sensitive memory show the most extensive overlap. Finally we note that, although APL:DPM gap junctions are implicated for ASM and for ethanol sensitivity, cell type-specific sequencing and immunohistology with antibodies fail to detect INX6:INX7 gap junctions in these neurons [95,96]. One possibility is that gap junctions play a developmental role that manifests in adult behavior. Alternatively, behavior may be a more sensitive readout for expression.

An outstanding question is whether ethanol memory-like states are associative or non-associative, or a mix thereof. Cues were not explicitly paired with the initiating ethanol exposures, and they were not used to test for tolerance. Hence, association, if occurring, is made to readily available cues in the environment, or perhaps even to interoception of the pharmacological action of ethanol. A single inebriating ethanol dose similar to that used herein to induce rapid tolerance also induces ethanol preference [97]. The ethanol preference test is performed in a different environment and at a different ethanol concentration than the initiating ethanol exposure, which ruled out many potential associative cues. Thus, at least some plasticity caused from the first ethanol exposure may be non-associative.

Our findings are limited in several ways. First, ethanol as a pharmacological agent has both positive and negative valence effects even within a single exposure. Here we compare ethanol responses to negative valence classical learning and memory, because current knowledge of positive valence classical learning and memory in *Drosophila* is more fragmentary. Second, as ethanol dose increases during a single exposure, animals proceed through a sequence of responses that likely engage partially overlapping sets of molecules and neurons. Our endpoint analysis of tolerance to the sedating effects of ethanol captures a specific aspect of ethanol action. Third, we limit our study to the APL, DPM, and Kenyon cells, whereas the classical learning and memory circuitry includes additional inputs and outputs, as well as neurons in other regions of the brain. The role of other known sites for rapid tolerance, including in the central complex, the blood-brain barrier, and parts of the circadian clock, were also not tested [77,98–100].

In conclusion, ethanol sensitivity and tolerance partially overlap molecularly and anatomically. The mushroom body circuitry is most intimately involved with initial ethanol sensitivity and AST, the shorter and labile form of rapid tolerance. Consolidated ART is largely or completely independent of the mushroom body circuitry studied here. The DPM neurons hold a protein synthesis dependent step for chronic tolerance, and therefore they may be a site for the encoding of the chronic memory-like state. Finally, regulation of Sirt1 by ethanol or experience may gate γ lobe Kenyon cell contribution to chronic tolerance development.

## Methods

### *Drosophila* culturing and strains

All strains used in this study were outcrossed for at least five generations to the Berlin genetic background carrying the $w^{1118}$ or $v^1$ marker mutation. The $w^{1118}$ Berlin strain served as a

control for loss-of-function mutants. Flies were cultured on standard cornmeal/molasses/yeast medium at 25°C and 60% relative humidity under a 12 h light/dark schedule. Flies for thermogenetic experiments were cultured the same, except at 18°C. All experiments used adult male flies that recovered from $CO_2$ collection at least 1 d before any behavioral paradigms. All *Drosophila* strains are listed in S1 Table.

## Ethanol behaviors

**Rapid tolerance.** Ethanol sensitivity and rapid tolerance were measured as previously described [30]. Briefly, groups of ~20 genetically identical flies (n = 1) were exposed to 100% humidified air or 55% ethanol vapor, an intermediate ethanol dose that induces submaximal rapid tolerance and leads to 50% sedation in 12–20 min in controls [75]. Flies were exposed to ethanol for 30 min during the first exposure (E1) to bring controls to 75–90% sedation. In rare cases where the experimental group exhibited strongly decreased sensitivity, the ethanol vapor concentration was increased to 60% and the E1 lengthened to 30–35 min. The time to 50% sedation (ST50) was calculated for each group (E1) based on the number of flies that lost the righting reflex at 6 min intervals. Flies were returned to standard housing, allowed to rest for 3.5 h or 23.5 h, then re-exposed to an identical concentration of ethanol vapor (E2). Rapid tolerance was calculated as the difference in ST50 between the two exposures, E2–E1.

**Chronic tolerance.** Flies housed in perforated 50 mL conical tubes with 5 mL of food were placed in a temperature- and light-controlled chamber for 48 h with continuous perfusion of either 100% humidified air or 16% ethanol vapor. After exposure, the flies were allowed to rest in the rearing incubator for 24–72 h. Chronic tolerance was measured between groups pre-exposed to ethanol minus those pre-exposed to air, with these ethanol and air pre-exposure group pairings randomly assigned within the daily cohort. A final n of 15–20 groups per treatment and genotype was achieved by repeating the experiment across multiple days.

## Cold shock anesthesia

A brief anesthetic cold shock was administered to flies either 30 min after E1, as previously documented, or 30 min before E2 [10]. Standard fly vials housing ~20 flies were placed in a 4°C ice bath for 3 min to achieve anesthetic cold shock. Observationally, flies quickly lose locomotion on ice then regain it upon return to 25°C.

## Experimental design and statistical analysis

For all experiments, the experimental manipulation was tested in the same session as the genetically matched or treatment-matched controls. Data was collected across multiple days with progeny from repeat parental crosses and collated together without between-day adjustments. Untransformed (raw) data were input into GraphPad Prism 10.4.1 and used for the following statistical analyses: unpaired t test or Mann-Whitney test if F test revealed unequal variances, one-sample t test, one-way ANOVA followed with Holm-Šídák's multiple comparison test for normally distributed data, Kruskal–Wallis test followed with Dunn's multiple comparison test for data that fails the Shapiro-Wilk normality of residuals test, and Brown–Forsythe test with Dunnett's T3 post hoc test for data that fails the Brown-Forsythe test for equal standard deviations. The full statistical test results are reported in S2 Table. For experiments designed with more than one control, statistical significance is only interpreted when each control is different from the experimental. These interpretations are shown as significance indicators on the figures based on the results of t tests or ANOVA post hoc tests (****$p \leq 0.0001$; ***$p \leq 0.001$; **$p \leq 0.01$; *$p \leq 0.05$; and ns, $p > 0.05$). Error bars represent the SEM.

## Supporting information

**S1 Fig.** **A**) Ethanol sensitivity measurements for rapid tolerance cold shock tests. unpaired t-test (two tailed). **B**) 4 hr and 24 hr rapid tolerance in *rut¹* flies mutant for the Rutabaga adenylyl cyclase. Left: unpaired t test (two-tailed); middle: unpaired t test (two-tailed); right: unpaired t test (two-tailed). **C**) RNAi against *rsh* in all neurons decreases rapid tolerance without affecting sedation sensitivity. Tolerance: one-way ANOVA/Holm-Šídák's. Sensitivity: one-way ANOVA/Holm-Šídák's.
(EPS)

**S2 Fig.** **A**, **A'**) RNAi against *amn* in the mushroom body Kenyon cell γ lobes with a second *Gal4* driver reduces rapid tolerance and sedation sensitivity. Tolerance: one-way ANOVA/Holm-Šídák's. Sensitivity: one-way ANOVA/Holm-Šídák's. **B**, **B'**) RNAi against *CaMKII* in all mushroom body Kenyon cells has no effect on rapid tolerance or sedation sensitivity (left). Pan-neuronal *CamKII* RNAi specifically reduces rapid tolerance. Tolerance, left: one-way ANOVA; right: one-way ANOVA/-Holm-Šídák's. Sensitivity, left: one-way ANOVA; right: one-way ANOVA/Holm-Šídák's.
(EPS)

**S3 Fig.** **A**, **A'**) A second RNAi against *rsh* in all mushroom body Kenyon cells does not affect rapid tolerance, whereas it reduces sedation sensitivity. Tolerance, left: one-way ANOVA; right: one-way ANOVA. Sensitivity: Brown-Forsythe/Dunnett's T3. **B**, **B'**) A second RNAi against *rsh* in the APL neurons does not affect rapid tolerance or sedation sensitivity. Tolerance: one-way ANOVA. Sensitivity: one-way ANOVA. **C**) Expression pattern of *APL^VT-Gal4* driver *VT43924*, detected with myristoylated GFP and counterstained with anti-Bruchpilot (BRP). **D**) Expression pattern of *DPM^VT-Gal4* driver *VT64246*, detected with myristoylated GFP and counterstained with anti-Bruchpilot (BRP).
(EPS)

**S4 Fig.** **A**, **A'**) Neuronal inactivation of the APL neurons throughout the rapid tolerance paradigm, or only during tolerance acquisition, reduces rapid tolerance. Tolerance, left: Kruskal-Wallis/Dunn's; right: one-way ANOVA/Holm-Šídák's. Sensitivity, left: one-way ANOVA; right: one-way ANOVA/Holm-Šídák's. **B**, **B'**) RNAi against *Gad1* in the APL neurons using *GH146-Gal4* increases rapid tolerance (left), and the effect is sensitive to cold shock (right). Sedation sensitivity is reduced. Tolerance, left: one-way ANOVA/Holm-Šídák's; right: one-way ANOVA/Holm-Šídák's. Sensitivity: Brown-Forsythe/Dunnett's T3. **C**) RNAi against *Tbh* in the APL neurons has no effect on 24 hr rapid tolerance. One-way ANOVA. **D**, **D'**) Inactivation of the DPM neurons has no effect on rapid tolerance or sedation sensitivity. Tolerance, left and right: one-way ANOVA. Sensitivity, left: one-way ANOVA/Holm-Šídák's; right: one-way ANOVA. **E**, **E'**) Reducing GABA synthesis in the DPM neurons has no effect on rapid tolerance, but it decreases sedation sensitivity. Tolerance: one-way ANOVA. Sensitivity: Kruskal-Wallis/Dunn's. **F**, **F'**) Gap junctions between the APL and DPM neurons are dispensable for rapid tolerance, but they are required to promote sedation sensitivity. Tolerance, left: one-way ANOVA; right: one-way ANOVA. Sensitivity, left: one-way ANOVA/Holm-Šídák's; right: Kruskal-Wallis/Dunn's.
(EPS)

**S5 Fig.** **A**, **A'**) Synthesis of serotonin (*Ddc*: Dopa decarboxylase) in the DPMs is not required for chronic tolerance. Tolerance: one-way ANOVA/Holm-Šídák's. Sensitivity: Kruskal-Wallis/Dunn's. **B**, **B'**) *Rdl* RNAi in a second αβ driver, *17d-Gal4*, reduces chronic tolerance and sedation sensitivity. Tolerance: one-way ANOVA/Holm-Šídák's. Sensitivity: one-way ANOVA/Holm-Šídák's. **C**, **C'**) The *Rdl* GABA_A receptor does not function in the APL or DPM neurons

for chronic tolerance. Tolerance, left: Brown-Forsythe; right: one-way ANOVA. Sensitivity, left: Brown-Forsythe/Dunnett's T3; right: Kruskal-Wallis/Dunn's.
(EPS)

**S1 Table. *Drosophila* strains.**
(DOCX)

**S2 Table. Genotypes, numbers of trials, and statistical analyses for all experiments.**
(XLSX)

**S3 Table. Source data for all figures and graphs.**
(XLSX)

## Author contributions

**Conceptualization:** Caleb Larnerd, Fred W. Wolf.

**Data curation:** Caleb Larnerd, Fred W. Wolf.

**Formal analysis:** Caleb Larnerd, Fred W. Wolf.

**Funding acquisition:** Fred W. Wolf.

**Investigation:** Caleb Larnerd, Maria Nolazco, Ashley Valdez, Vanessa Sanchez.

**Methodology:** Caleb Larnerd.

**Project administration:** Fred W. Wolf.

**Writing – original draft:** Caleb Larnerd, Fred W. Wolf.

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
