## [Decision Letter · Decision Letter 0]

22 Nov 2024

PGENETICS-D-24-01147Memory-like states created by the first alcohol experience are encoded into the Drosophila mushroom body learning and memory circuitry in an alcohol-specific mannerPLOS GeneticsDear Dr. Wolf, Thank you for submitting your manuscript to PLOS Genetics. It was an interesting read!  After careful consideration, the reviewers and Associate Editor feel that it is appropriate for PLOS Genetics but would benefit from minor revisions. Therefore, we invite you to submit a revised version of the manuscript that addresses the points raised during the review process. Please submit your revised manuscript, preferably within 30 days (but this is flexible). If you need more time, please reply to this message or contact the journal office at plosgenetics@plos.org. Please include the following items when submitting your revised manuscript:

* A rebuttal letter that responds to each point raised by the editor and reviewer(s). You should upload this letter as a separate file labeled 'Response to Reviewers '. This file does not need to include responses to formatting updates and technical items listed in the 'Journal Requirements' section below.

* A marked-up copy of your manuscript that highlights changes made to the original version. You should upload this as a separate file labeled 'Revised Manuscript with Track Changes '. * An unmarked version of your revised paper without tracked changes. You should upload this as a separate file labeled 'Manuscript '. If you would like to make changes to your financial disclosure, competing interests statement, or data availability statement, please make these updates within the submission form at the time of resubmission. Guidelines for resubmitting your figure files are available below the reviewer comments at the end of this letter. We look forward to receiving your revised manuscript. Kind regards,

Julie Simpson

Academic Editor

PLOS Genetics

Anne O'Donnell-LuriaSection EditorPLOS Genetics

Aimée Dudley

Editor-in-Chief

PLOS Genetics

Anne Goriely

Editor-in-Chief

PLOS Genetics

**Additional Editor Comments:**

As a bystander to this field, I do encourage you to consider including some additional, brief general context in the introduction or discussion.  That will make it easier for more readers to appreciate the significance of the parallels between associative learning and response to ethanol exposure... I understood this most clearly from the cover letter, so perhaps some of that text could be added.  The proposal that ethanol exposure could interfere with memory formation because of the shared circuitry/molecular mechanisms is intriguing - was this tested?  I look forward to reviewing the revised version and hope to see your work in press soon. 

**Journal Requirements:**

4) We notice that your supplementary Figures, and Tables are included in the manuscript file. Please remove them and upload them with the file type 'Supporting Information'. Please ensure that each Supporting Information file has a legend listed in the manuscript after the references list.

**Reviewers' comments:**

**Reviewer #1:**   Larnerd et al., describe in their manuscript ”memory-like states created by the first alcohol experience are encoded into the Drosophila mushroom body learning and memory circuitry an alcohol-specific manner” whether the formation of rapid and chronic tolerance is sensitive to a cold shock and what kind of neurons might regulate cold shock sensitive rapid tolerance, cold chock insensitive rapid tolerance or chronic tolerance.

It is important to classify the mechanisms underlying rapid and chronic tolerance into anesthesia-sensitive and anesthesia-resistant mechanisms. It is also important to map neurons that contribute to rapid and chronic tolerance. In addition, Larnerd et al. have begun to address whether similar molecules underlying the effects of ethanol at the cellular level also contribute to rapid and chronic tolerance in the neurons identified.

What is misleading is the direct comparison to mechanisms underlying aversive associative learning and memory.

First, the effect of ethanol is doses dependent. We do not know whether the SD50 of the first ethanol exposure in flies is evaluated as negative or positive. Positive and negative effects might contribute to the development of tolerance. The ethanol exposure might result in a mixture of effects- either positive or negative.

Second, the endpoint of E1 (is described in the material and method as “between 12 to 20 min”) is very unprecise. If the endpoint was fixed, the question arises whether the effect for the flies is either negative or positive. That will influence whether the mechanism of rapid tolerance to the sedative effect of ethanol are compared to mechanism underlying appetite or aversive olfactory learning and memory.

What would be okay, is to use aversive olfactory learning and memory together with the mushroom body as working model. That would required to have detailed introduction to the function of DPM and Apl neurons in different forms of memory in addition to a description what part of the MB contribute to what kind of memory. Here, an introductory figure would be great. We also know that a single exposure to ethanol changes the function of dopaminergic neurons that mediate the positive reinforcing effect of ethanol (Knabbe et al., 2022). A comparison to mechanisms underlying appetitive olfactory learning and mechanism might be more interesting. Given that aversive and appetite memory requires different groups of dopaminergic neurons, I would highlight one model (aversive or appetitive).

In general, the experiments were carried out carefully and I think the data are important enough to be published in Plos Genetics. However, I recommend major revision before publishing.

In detail:

1) Learning and memory in animal models are defined by observation. An animal learns when after a previous experience a change in behavior is observed. If the behavior change persists, it is due to the memory of the previous experience. I am not sure how the term “memory like state” relate to this. Per definition ethanol tolerance is learnt and memorized (see definition above). There is literature related to this, that has not been cited and need to be included into the introduction. That might serve for a better introduction and understanding.

Nevo and Hamon, 1995

Kalant , 1998

Bitrán M, Kalant H, 1991. Learning factor in rapid tolerance to ethanol-induced motor impairment. Pharmacology, Biochemistry, and Behavior 39, 917–922.

There are two aspects to ethanol: the direct action of the drug on the cell and the association of the resulting intoxication with the environment that might influence the behavioral outcome.

In addition, the reinforcing actions of drugs of abuse is either positive or negative depending on the concentration and duration of ethanol intoxication.

Barreto PS, Lemos T, Morato GS, 1998. NMDA-receptor antagonists block the development of rapid tolerance to ethanol in mice. Addiction Biology 3, 55–64. 10.1080/13556219872344 - DOI - PubMed

2) The work would benefit from background information related learning and memory and its relationship to ethanol tolerance (mouse, rats see references see above).

3) The work would benefit from background from molecular mechanism under ethanol tolerance (GABA, glutamate balance…).

4) Reorganization of the data. Usually it is important to know if the controls work and then see the experimental data. Not the other way around. We also read scientific articles from left to right and not right to left.

5) The statistics: there is no indication of the sample size. I am not willing to count dots. The correction of significance using Dunett’s test does not appear correct.

This is a test for multiple comparisons with a control are also called many-to-one comparisons. That means to one control. In Figure 1D. left panel, the data are compared to one control, however in addition to each other. The data in 1A third panel 24h memory, there is barely any memory observed. Is the E2 different from E1?

6) In Figure 1A, the data are presented in the correct order, first “normal” than treatment group. Please resort data presentation

7) The data of figure 10 does not really fit to the story and should be removed.

Minors:

Title: Alcohol should be ethanol

Introduction:

Line 67: Apl and DPM are abbreviations

Line 74: Reference is missing.

Line 76 to 77: reference is missing.

Results:

Line: it is not clear how ethanol tolerance is measured. First dose how long? Second dose how long? Info is also missing in material and method section. Brief introduction how experiments were done is missing.

Line 87: reference for ASM and ARM is missing

Question: If amn is required for in the mushroom body for AST, how that can be connected to the GABA function (Fig5) that repress Kenyon cells for AST and NMDar1 knockdown in mb cells?

Line 115 to 116 : appears redundant to Line 118 to 121

Line 153: awkward sentence: It also mapped…

Line 163: reference is missing for shi

Line 207: Thus, activity of the GABA ergic Apl neuron… too strong statement related to DPM neuron function. The experiment DPM, UAs-Shi in Ast was not done or GAD knock down… maybe I did not see it.

Line 276: difficult to evaluate, increase number of N or down tune…how much N anyway?

Section of Sirt 1: I am not sure how this add up to the rest of the story, especially in the light of reference 47 showing that Sirt1 terminates Mef2 induction.

Fig2: The model in E: if the neurons are not addressed or do not have a function, they should be labeled weakly or removed. It is also not clear what the arrow and cold shock means in this context and why this is not shown vor amn in the memeory model”.

Similar comments apply also for the other cartoon in Figure 3 an there on. Too much infos in a model without relationship to text blur content of model. It might be okay in a final summary model to include all infos.

Please change the order of the controls and the experimental groups, otherwise it is very hard to follow.

Discussion:

- I missed something how molecules/brain region relate to infos known about rapid tolerance in mice.

- In addition, I think it is too simple to look at ethanol exposure as “one” behavior. For example, sedation might have different valence to flies than hyperactivity.

- Rapid tolerance requires hang function in neurons of the central complex. Homer is also required in the central complex for tolerance. How does this relate to a possible function outside of the MB for ART?

**Reviewer #2: **  Larnerd et al. (2024) investigates the how “memory”-like ethanol tolerance states are encoded in the fly brain. Through procedural and careful molecular and circuit analyses they identify a set of neurons and molecules necessary for different forms of ethanol tolerance and sensitivity. This manuscript provides interesting comparisons between ethanol tolerance and memory; both involve mushroom body circuitry and share interesting similarities in the timing of their involvement throughout “acquisition” and “consolidation”.

Although the authors compare these forms of tolerance to memory-like states, they do not explicitly state what this association could be (what constitutes the conditioned and unconditioned stimuli?). While this is briefly mentioned in the discussion (Lines 413 – 423) it would be useful to make it clearer what this possible association could be – or if the authors believe there is no explicit association being made stating this in the text. It seems like the authors suggest that the conditioned stimulus could be the odor of ethanol. In this case, it would be interesting to test smell-blind flies (e.g. olfactory mutants or shibire-ts olfactory sensory neurons) to determine if their ability to acquire tolerance depends on their ability to smell the ethanol. 

It is not clear if the molecular targets of interest are indeed expressed in the cells the authors target for manipulation. I would recommend the authors look to Aso et al. (2019) (Figure 3 and supplements) and the FlyCellAtlas (Li et al. 2022) to compare/confirm gene expression for those cell types and adjust the wording in the paper taking into consideration this data, for those that show no expression of the genes of interest in those cell types. i.e. gap junctions inx6 and inx7 don’t appear to be clearly expressed in APL & DPM^[3]^ , which could explain why the authors did not see a phenotype for the rapid tolerance. 

Minor Comments 

If blocking synaptic transmission of APL reduces rapid tolerance (Fig 4A) and knock-down of GABAAR in KCs also suppresses tolerance (Fig 8B), does it make sense that suppressing GABA synthesis (Fig 4D) would enhance rapid tolerance? In Figure 1A (third and fourth panels) Why are the values for the untreated groups so different between the two panels? It looks like they have not acquired any tolerance in the 4^th^  panel despite having the same treatment as the untreated flies in the 3^rd^  panel. I suggest changing the colors for rapid tolerance so that it is more easily distinguishable from chronic tolerance. Some of the “Fig X” are missing a dot between Fig and X. E.g. Line 154 (Fig 3B’). Figure S1 has panels with bars occluding the dots and error bars. 

[1] Aso, Y., Ray, R. P., Long, X., Bushey, D., Cichewicz, K., Ngo, T. T., ... & Rubin, G. M. (2019). Nitric oxide acts as a cotransmitter in a subset of dopaminergic neurons to diversify memory dynamics. *Elife* , *8* , e49257.

[2] Li, H., Janssens, J., De Waegeneer, M., Kolluru, S. S., Davie, K., Gardeux, V., ... & Wolfner, M. F. (2022). Fly Cell Atlas: A single-nucleus transcriptomic atlas of the adult fruit fly. *Science* , *375* (6584), eabk2432.

[3] Ammer, G., Vieira, R. M., Fendl, S., & Borst, A. (2022). Anatomical distribution and functional roles of electrical synapses in Drosophila. *Current Biology* , *32* (9).

**Reviewer #3:**  In their manuscript entitled “Memory-like states created by the first alcohol experience are encoded into the Drosophila mushroom body learning and memory circuitry in an alcohol-specific manner,” Larnerd et al. extend their previous work on rapid and chronic teolerance states by using cold shock to further define consolidated processes underlying tolerance. They define new molecular and circuit-level events by drawing further parallels between memory states and tolerance. The work is comprehensive, novel, and welcome, as it should significantly impact and integrate how we understand rapid and chronic tolerance in Drosophila.

There are minor changes that I believe will help readers:

In the methods section, please specify how long flies were expose to ethanol vapor during the first exposure. I infer this is 30 minutes, but it is not obvious and readers should not need to go to previous papers for this infomration.

The w1118 mutation was used as a control for the loss-of-function mutants. Given that white mutations have been reported to lead to defects in ethanol sensitivity (Chan et al. 2014; van der Linde, et al. 2014), the authors should address this and how it may or may not impact the analysis of the loss-of-function mutants. Since these mutant phenotypes were also supported by RNAi studies, this is not a major concern.

In the Introduction, the authors refer to the hundreds of dopaminergic neurons that innervate the mushroom bodies (page 4, line 72). This reviewer is aware of ~125 neurons innerating the MB per hemisphere: PAMs and PPL1s (i.e., Tanaka , et al. 2008; Mao and Davis, 2009). It would be helpful if the authors were a more precise here.

The role of Sirt1 is left out of Figure 11, which otherwise summarizes the findings in the maunscript.

**Figure resubmission:**

While revising your submission, please upload your figure files to the Preflight Analysis and Conversion Engine (PACE) digital diagnostic tool, https://pacev2.apexcovantage.com/. PACE helps ensure that figures meet PLOS requirements. To use PACE, you must first register as a user. Registration is free. Then, login and navigate to the UPLOAD tab, where you will find detailed instructions on how to use the tool. If you encounter any issues or have any questions when using PACE, please email PLOS at figures@plos.org. Please note that Supporting Information files do not need this step. If there are other versions of figure files still present in your submission file inventory at resubmission, please replace them with the PACE-processed versions.**Reproducibility:**

---

## [Editor Report · Decision Letter 1]

17 Jan 2025

Dear Dr Wolf,

We are pleased to inform you that your manuscript entitled "Memory-like states created by the first ethanol experience are encoded into the Drosophila mushroom body learning and memory circuitry in an ethanol-specific manner" has been editorially accepted for publication in PLOS Genetics. Congratulations!

Yours sincerely,

Julie Simpson

Academic Editor

PLOS Genetics

Gregory Cooper

Section Editor

PLOS Genetics

Aimée Dudley

Editor-in-Chief

PLOS Genetics

Anne Goriely

Editor-in-Chief

PLOS Genetics

Comments from the reviewers (if applicable):

Dear Dr. Wolf and co-authors,

Thank you for your conscientious response to the reviewers' recommendations. PLOS Genetics is pleased to accept the revised and improved manuscript. Congratulations on your work.

**Data Deposition**

http://datadryad.org/submit?journalID=pgenetics&manu=PGENETICS-D-24-01147R1

**Press Queries**

---

## [Editor Report · Acceptance letter]

PGENETICS-D-24-01147R1

Memory-like states created by the first ethanol experience are encoded into the Drosophila mushroom body learning and memory circuitry in an ethanol-specific manner

Dear Dr Wolf,

We are pleased to inform you that your manuscript entitled "Memory-like states created by the first ethanol experience are encoded into the Drosophila mushroom body learning and memory circuitry in an ethanol-specific manner" has been formally accepted for publication in PLOS Genetics! Your manuscript is now with our production department and you will be notified of the publication date in due course.

With kind regards,

Zsofia Freund

PLOS Genetics

On behalf of:
